# Single-cell profiling identifies a CD8[bright] CD244[bright] Natural Killer cell subset that reflects disease activity in HLA-A29-positive *birdshot chorioretinopathy*

Pulak R. Nath [1,6] ✉, Mary Maclean[1,7], Vijay Nagarajan [1,2], Jung Wha Lee[1], Mehmet Yakin[1], Aman Kumar[1], Hadi Nadali[1], Brian Schmidt[3], Koray D. Kaya[4], Shilpa Kodati[1], Alice Young[3], Rachel R. Caspi [2], Jonas J. W. Kuiper [5,8] ✉ & H. Nida Sen[1,8]

Birdshot chorioretinopathy is an inflammatory eye condition strongly associated with MHC-I allele HLA-A29. The striking association with MHC-I suggests involvement of T cells, whereas natural killer (NK) cell involvement remains largely unstudied. Here we show that HLA-A29-positive birdshot chorioretinopathy patients have a skewed NK cell pool containing expanded CD16 positive NK cells which produce more proinflammatory cytokines. These NK cells contain populations that express *CD8A* which is involved in MHC-I recognition on target cells, display gene signatures indicative of high cytotoxic activity (*GZMB*, *PRF1* and *ISG15*), and signaling through NK cell receptor CD244 (*SH2D1B*). Long-term monitoring of a cohort of birdshot chorioretinopathy patients with active disease identifies a population of CD8[bright] CD244[bright] NK cells, which rapidly declines to normal levels upon clinical remission following successful treatment. Collectively, these studies implicate CD8[bright] CD244[bright] NK cells in birdshot chorioretinopathy.

Non-infectious uveitis (NIU) is a clinically and prognostically heterogeneous group of ocular inflammatory diseases and a major cause of severe visual handicap[1]. Birdshot chorioretinopathy (birdshot uveitis or BCR-UV) is a relatively rare form of NIU that has clinically distinct features in the form of retinal and choroidal inflammatory lesions visible on eye examination. BCR-UV can lead to progressive deterioration of visual function[2–4] due to persistent inflammation in the retina and choroid[5]. Consequently, patients often require systemic immunomodulatory therapy to control ocular inflammation[3,6]. The

disease mechanisms driving BCR-UV remain to be elucidated, but scientific advances have shed light on the immunopathology of this clinically well-defined ophthalmological condition[7].

One of the most striking molecular features of BCR-UV is that it's strongly associated with the presence of HLA-A29 allele such that all patients carry the HLA-A29 allele (most often the common *HLA-A\*29:02*)[7,8]. Genome-wide genetic studies have revealed that susceptibility to BCR-UV also maps to other factors of the MHC-I pathway[9–11]. This incriminates CD8+ T cells and Natural Killer (NK) cells in the

[1]Clinical and Translational Immunology Unit, Laboratory of Immunology, NEI, NIH, Bethesda, USA. [2]Immunoregulation Section, Laboratory of Immunology, NEI, NIH, Bethesda, USA. [3]NIH Intramural Sequencing Center, NIH, Rockville, USA. [4]Medical Genetics and Ophthalmic Genomics Unit, NEI, NIH, Bethesda, USA. [5]Department of Ophthalmology, University Medical Center Utrecht, University of Utrecht, Utrecht, Netherlands. [6]Present address: Lentigen Technology Inc., A Miltenyi Biotec Company, 910 Clopper Road, Gaithersburg, MD 20878, USA. [7]Present address: Translational Immunology Section, Office of Science and Technology, NIAMS, Bethesda, NIH, USA. [8]These authors jointly supervised this work: Jonas J. W. Kuiper, H. Nida Sen. ✉e-mail: hellopran2000@gmail.com; j.j.w.kuiper@umcutrecht.nl

pathogenesis of BCR-UV[7]. Indeed, CD8[+] T cells have been shown to infiltrate eye tissues of patients and secrete cytokines and is considered central to its immunopathology[12–15].

The role of NK cells in BCR-UV has remained significantly underexplored. NK cells are the major innate lymphoid cells (ILC) that mediate cytotoxicity as well as immune-modulation by producing significant amounts of inflammatory cytokines including interferon-gamma (IFN-γ) and tumor necrosis factor-alpha (TNF). NK cells can be broadly divided into two categories; conventional NK cells (cNK) are located in bone marrow and in peripheral blood and the tissue-resident NK cells (trNK) that are found in lymph nodes, the thymus, lung, liver, small intestine, and in the uterus. trNK cells are considered developmentally and functionally distinct from cNKs.

Classically, cNK cells are subdivided into two well-established overarching populations; the CD56[dim] CD16[+] NK cells (~90% of circulating NK cells) which are considered to be cytotoxic and also produce pro-inflammatory cytokines and the CD56[bright] (~10% of circulating NK cells) that are considered to be more immunoregulatory[16]. Of interest, CD56[bright] NK cells have been shown to be lower in patients with NIU, including BCR-UV patients with active uveitis[17] and successful immunosuppressive treatment of NIU is accompanied by the recovery of CD56[bright] NK cell levels[18].

Beyond this phenotypic bifurcation, the NK cell population is considerably more diverse, with at least 10 transcriptionally distinct clusters in peripheral blood identified through single-cell RNA sequencing (scRNAseq)[19–21]. This includes CD56[dim] CD16[+] NK subsets which may be 'inflammatory', defined by high levels of expression of cytokine and interferon response genes[20,21] and "adaptive-like" NK populations that expand during infection[22]. Transcriptomically distinct NK cell subsets are considered to exhibit differential effector functions mediated by an ensemble of surface immunoregulatory molecules[23], in particular Killer cell immunoglobulin-like receptors (KIRs), IgG Fc receptors (e.g., CD16), and integrins (e.g., CD47)[24–26]. Perturbations in the composition of the NK cells have been reported in other MHC-I associated conditions, such as HLA-B27-positive *ankylosing spondylitis*[27] and were shown to be predictive for clinical outcome in autoimmune diseases, such as Multiple Sclerosis[28]. Collectively, these observations suggest that deep phenotyping of the blood NK cell compartment could provide better understanding of disease biology and may contain relevant information about the clinical course of BCR-UV.

Here, we phenotype the NK cell compartment at single-cell resolution of patients with BCR-UV and report on the expansion of a CD56[dim] CD16[+] subset of NK cells which are CD8[bright] and CD244[bright], and whose reduction is correlated with clinical improvement after systemic immunosuppressive therapy.

## Results
### Increased frequency of CD56[dim]CD16[+] NK cells in birdshot uveitis patients

In order to determine whether NK cells contribute to the disease mechanisms of BCR-UV, we first looked for evidence that circulating NK cells were specifically affected. To this end, we quantified the major lineages of immune cells (12-marker panel, 6 lineages; Supplementary Fig. 1A) in peripheral blood using flow cytometry in a cohort of 139 non-infectious uveitis (NIU) patients (including 18 BCR-UV patients) and 80 healthy controls (HC), (Fig. 1A). Global comparison of major lineages in all NIU patients versus healthy controls revealed a significant increase in frequency of blood NK cells (Fig. 1B), but not T cells, B cells, monocytes, or dendritic cells (Supplementary Fig. 1B–D). Flow cytometry analysis revealed that the frequency of blood NK cells appeared to be increased in several uveitis subtypes (Serpiginous, Definite Sarcoidosis and Birdshot), but this increase was most significant for BCR-UV (Birdshot, $P < 0.0001$; Fig. 1C). This became more evident after quantification of the two major.

NK populations that can be distinguished by their expression of surface CD56 and CD16 (i.e., CD56[bright] and CD56[dim]; Fig. 1D). We detected a significant increase of CD56[dim] CD16[+] cells and a concomitant decrease of the CD56[bright] CD16[-] cells only in BCR-UV patients or when considering all NIU patients collectively, but not individually in any of the other types of NIU (Supplementary Fig. 1E, F). This skew in CD56[dim]/CD56[bright] balance also remained evident after strict comparison to 15 age-matched healthy controls (mean age±SD = 62.2 ± 8.8; Fig. 1E).

Importantly, NK cells of BCR-UV patients showed enhanced responsiveness to restimulation by production of significantly higher tumor necrosis factor-α (TNF, $P = 0.007$) and interferon-γ (IFN-γ, $P = 0.002$; Fig. 1F), indicating that the altered CD56[dim]/CD56[bright] balance results in a more pro-inflammatory NK repertoire. CD56[bright] NK cells have been reported to be potent cytokine-secreting cells[29]. We compared IFN-γ and TNF production by CD56[bright] and CD56[dim] populations after stimulation with a leukocyte activation cocktail. This analysis revealed that CD56[bright] and CD56[dim] NK population produced comparable levels of IFN-γ and TNF in response to stimulation (Supplementary Fig. 1G). Moreover, CD56[dim] NK cells from uveitis patients did not produce significantly more of these cytokines compared to the controls. Collectively, these data show an imbalance in CD56[dim]/CD56[bright] cells in the peripheral blood of patients with BCR-UV and a skew towards a more proinflammatory phenotype that is not merely the result of CD56[dim]/CD56[bright] balance. The polyclonal stimulation did not affect the production of cytokines by CD56[bright] and CD56[dim] NK cells. However, these subsets may differ depending on the type of stimulation used.

### PBMC scRNA-seq identifies altered NK repertoire in birdshot chorioretinopathy

To allow characterization of the changes in peripheral blood NK cells in BCR-UV in an unbiased manner, we used single-cell RNA-sequencing (scRNAseq) of peripheral blood mononuclear cells (~300 K PBMCs) of 12 BCR-UV patients and 12 healthy controls (Fig. 2A and Supplementary Fig. 2A). Unsupervised clustering followed by uniform manifold approximation and projection (UMAP) and automated cell type annotation, identified an NK cell population (9,619 cells of cluster C4_Natural Killer cells, Fig. 2B and Supplementary Fig. 2B) with an altered NK cluster structure in two-dimensional UMAP space in BCR-UV patients compared to healthy controls (Supplementary Fig. 2C). NK-specific *GZMB* (granzyme B), *KLRF1* (NKp80), *KLRD1* (CD94), *GNLY* (Granulysin), *PRF1* (perforin), *NKG7* (Natural Killer Cell Granule Protein 7), SH2D1B (Ewing's sarcoma-activated transcript-2 or EAT2, CD244 signaling) and *GZMA* (granzyme A) were among the most differentially upregulated genes in BCR-UV (Supplementary Fig. 2D). We extracted scRNAseq data for cells corresponding to NK cells for further analysis. Unsupervised clustering of the NK cell population revealed a high level of transcriptomic heterogeneity and the existence of 12 distinct clusters ranging from 57 cells (cluster 11) to 2,093 cells (cluster 0, Fig. 2C and Supplementary Fig. 3A). Gene expression levels of characteristic NK lineage surface markers revealed that these clusters expressed different levels of transcripts encoding NK activating receptors *CD244* and *CD8A*, as well as NK inhibitory receptors *KIR3DL1*, *KLRD1* and *B3GAT1* (Fig. 2D) which is compatible with an altered composition of functional NK subsets. At a false discovery rate of 5%, cluster 1 was significantly decreased (948 cells in HC vs 275 cells in BCR-UV) while clusters 2 (351 vs 862 cells in HC vs BCR-UV), 6 (208 vs 457 cells in HC vs BCR-UV) and 10 (25 vs 235 cells in HC vs BCR-UV) were significantly increased in frequency in BCR-UV patients compared to healthy controls (Fig. 3A). Clusters 1, 2, 6 and 10 were uniquely represented by the expression of *MYOM2*, *SH2D1B*, *IGFBP7* and *LINC00996* genes, respectively (Supplementary Fig. 3B). We used the ABIS database to check the expression of least known *MYOM2*, *IGFBP7* and *LINC00996* genes across several

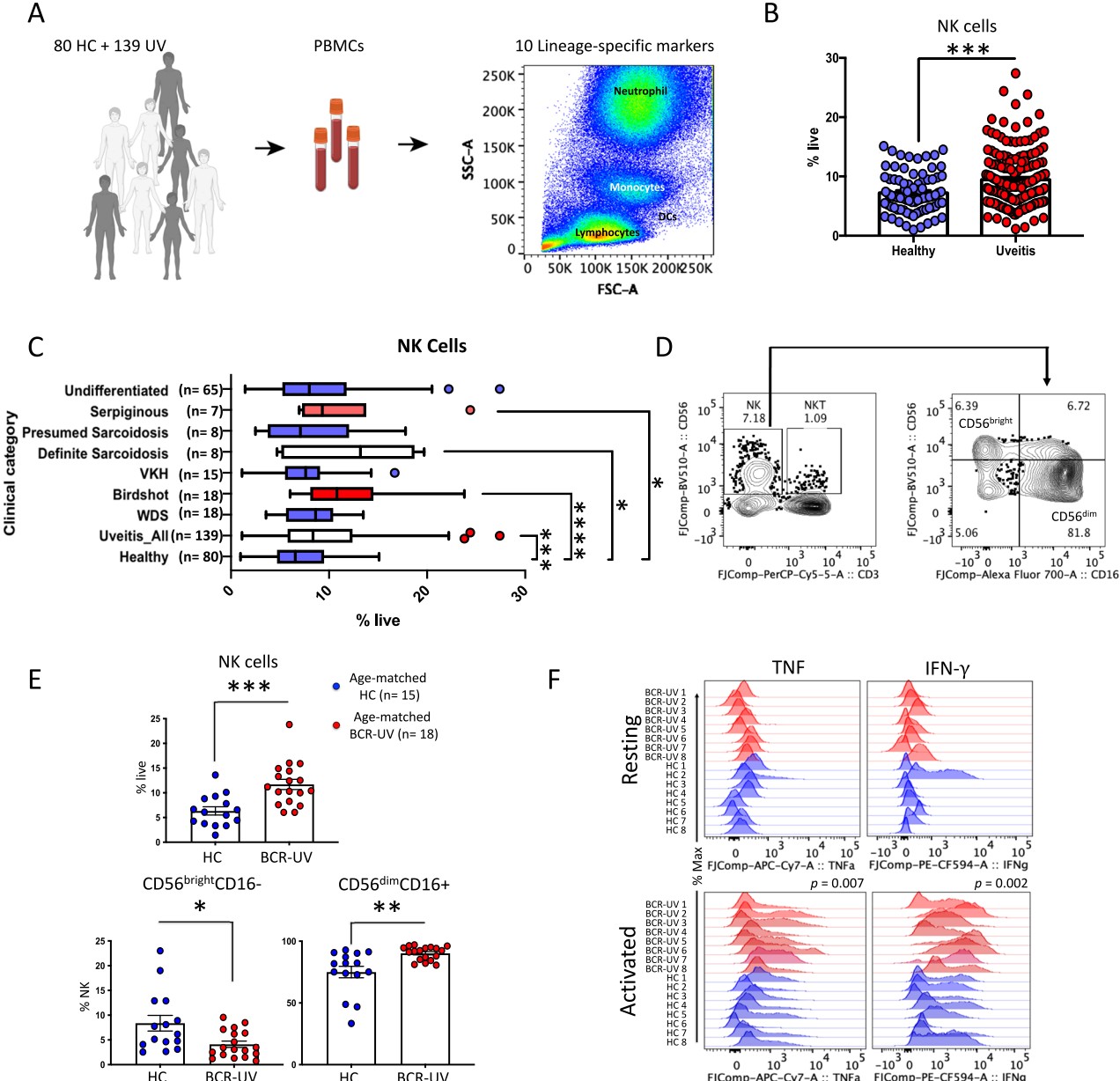

**Fig. 1 | Flow-cytometry profiling revealed altered Natural killer (NK) cell composition in peripheral blood of birdshot uveitis. A** Schematic representation of flow cytometry of major immune cell lineages in fresh peripheral blood of 139 uveitis patients (UV) and 80 age-, sex-matched healthy controls (HC). Figure is created with BioRender.com (www.biorender.com), released under a Creative Commons Attribution-NonCommercial-NoDerivs 4.0 International license. **B** Flow-cytometry quantification of the percentage of NK cells in the peripheral blood of uveitis patients (Uveitis) compared to healthy controls (Healthy). Healthy $n = 80$; Uveitis $n = 139$. ***$P = 0.0006$. Data are presented as mean values ± SEM. Source data are provided as a Source Data file. **C** Flow-cytometry analysis of NK cells (CD3−CD19−CD56+) in the fresh blood of different uveitis subgroups indicate the significant expansion of NK cells were restricted to birdshot, definite sarcoidosis and serpiginous sub-groups of uveitis cohort. Values are presented in the form of box and whiskers plot and represent medians with ranges (Whiskers: two-sided Tukey test). Healthy $n = 80$; Uveitis $n = 139$; WDS $n = 18$; Birdshot $n = 18$; VKH $n = 15$; Definite Sarcoidosis $n = 8$; Presumed Sarcoidosis $n = 8$; Serpiginous $n = 7$;

Undifferentiated $n = 65$. P values are from non-parametric Mann–Whitney U test, ****$P < 0.0001$, ***$P = 0.0006$, **$P = 0.004$, *$P = 0.02$. WDS, White Dot Syndromes; VKH, Vogt–Koyanagi–Harada disease. Source data are provided as a Source Data file. **D** The flow-cytometry gating strategy for CD56bright and CD56dim subsets of NK cells using CD56 and CD16 in peripheral blood. **E** NK cell and CD56bright and CD56dim subset quantification in peripheral blood of birdshot uveitis (BCR-UV) and age-matched healthy controls (HC). HC $n = 15$; BCR-UV $n = 18$. P values are from two-sided unpaired t test, ***$P = 0.0004$, **$P = 0.002$, *$P = 0.01$. Data are presented as mean values ± SEM. Source data are provided as a Source Data file. **F** The fluorescence intensity of TNF and IFN-γ produced by all NK cells, as determined by flow cytometry analysis upon stimulation with Lymphocyte Activation Cocktail (BD Biosciences). Analysis was conducted by stimulating PBMCs and flow-gating on NK cells from BCR-UV (red) and healthy controls (HC, blue). HC $n = 8$; BCR-UV $n = 8$. Statistical analysis was done using one-way ANOVA- Tukey's multiple comparisons test; P values are indicated above the histograms. Source data are provided as a Source Data file.

peripheral blood immune cell populations[30] (available via: https://giannimonaco.shinyapps.io/ABIS/). While *LINC00996* is expressed in NK cells at moderate levels, *MYOM2* and *IGFBP7* genes are highly expressed in the NK cells (Supplementary Fig. 3C).

Cluster 1 also showed high expression of *DUSP1, FOS, JUN,* and *CD69* (Fig. 3B) highly reminiscent of the gene expression profile of CD56bright CD16− NK cells[20], which corroborates our findings by major lineage flow cytometry. In contrast, the increased clusters 2, 6 and 10 expressed

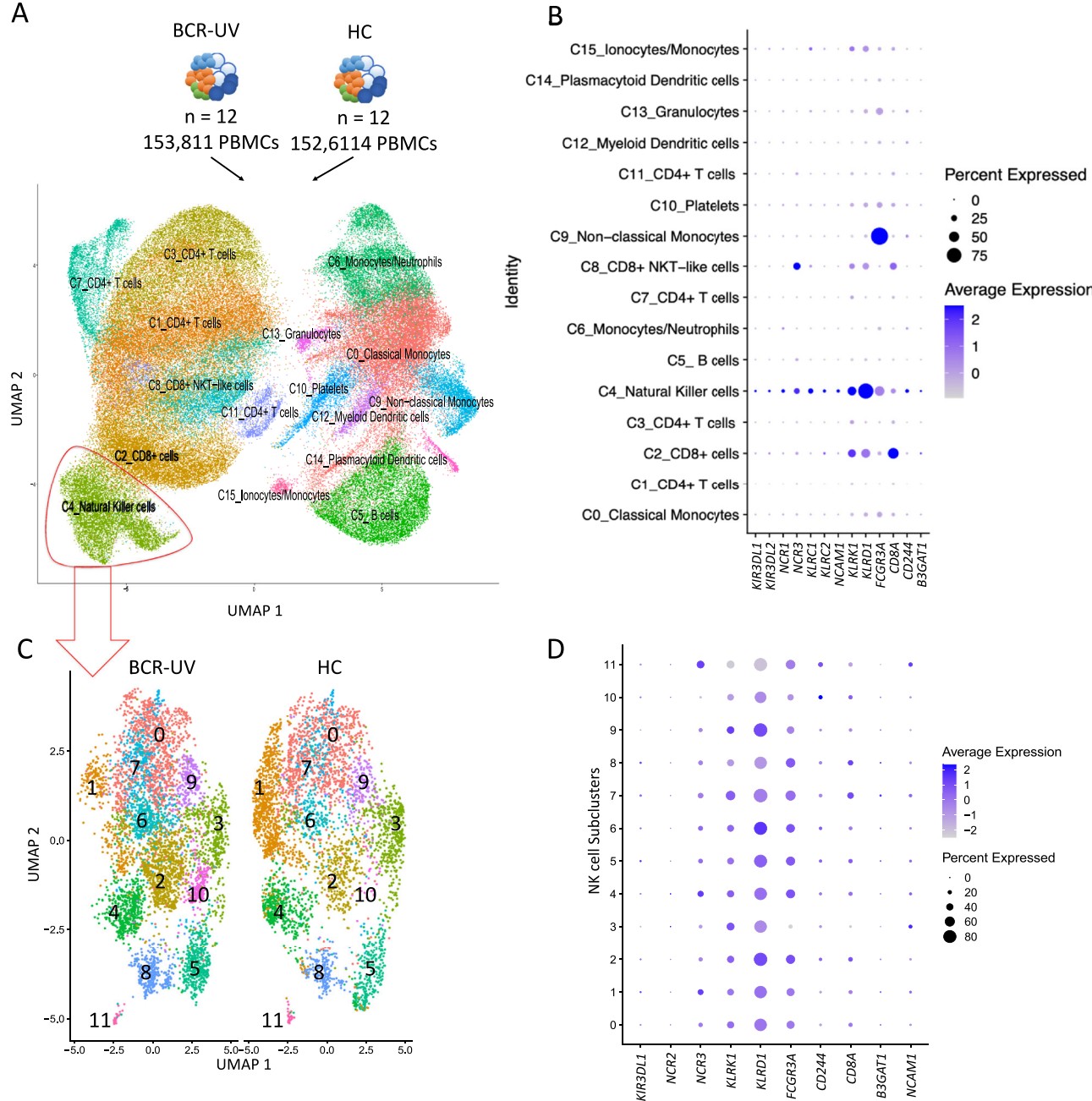

**Fig. 2 | Single-cell RNAseq analysis identifies transcriptional NK cell subsets.**
**A** UMAP plot of 306,425 PBMCs from 12 birdshot uveitis patients (BCR-UV) and 12 healthy controls (HC). The identified lineage clusters are highlighted (color-coded). **B** Dot plot showing the expression profile of NK surface marker-encoding genes across the annotated cell clusters identified in *A*. **C** UMAP plots of NK cells gated from the C4_Natural Killer cells cluster in *A* comparing NK subclusters in BCR-UV and HC. The 12 transcriptional NK cell subclusters are color-coded. **D** Dot plot showing the expression profile of NK surface marker encoding genes in each of the subclusters as identified in *C*.

lower levels of *NCAM1* (CD56; Fig. 2D) but high levels of *FCGR3A* (CD16) and the surface co-receptor encoding *CD8A* (CD8 Antigen, Alpha Polypeptide; Fig. 2D), which suggests that clusters 2, 6, 10 are subpopulations of the bulk population of CD8+ CD56^dim CD16+ NK cells.

We further observed that cluster 10 showed high *CD244* (Figs. 2D, 3B) and clusters 2 and 10 displayed enrichment of CD244 binding *Src homology 2* (SH2) domain-encoding genes *SH2D1B*, which control signal transduction through the surface receptor CD244[31] (Fig. 3B). This implicates active CD244 signaling in these NK clusters. Other highly expressed activation-associated genes in these sub-clusters include *SH2D2A* (T Cell-Specific Adapter Protein or TSAd), *TNFRSF18* (also known as *GITR*) and *ISG15* (Interferon-Stimulated Protein, 15 kDa; Fig. 3B,

Supplementary Fig. 3D). In summary, these results indicate expansion of activated CD8+ CD56^dim subpopulation characterized by high levels of CD244-signaling molecules in the blood of patients with BCR-UV.

**High-dimensional cytometry reveals accumulation of a CD8^bright CD244^bright subset of CD16+ NK cells in the circulation of birdshot chorioretinopathy patients**
We wished to validate our scRNAseq findings using a 12-marker panel flow cytometric phenotyping of the blood NK cell repertoire (Supplementary Fig. 4A). Unbiased cell clustering considering the surface marker phenotypes by FlowSOM discerned 12 NK cell clusters (Fig. 4A). As expected, the majority of the clusters were CD56^dim CD16+ (cluster 0-2, 5-

A

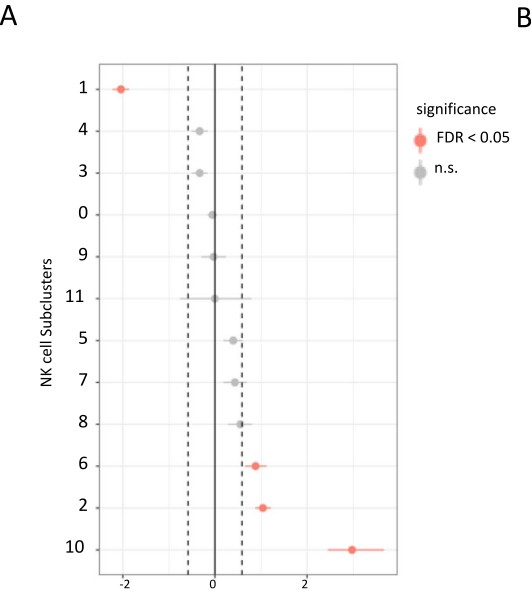

B

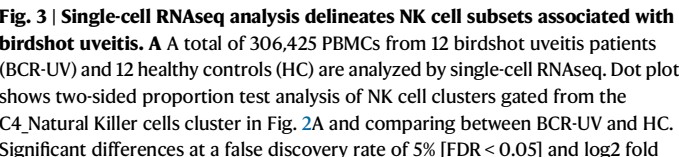

**Fig. 3 | Single-cell RNAseq analysis delineates NK cell subsets associated with birdshot uveitis. A** A total of 306,425 PBMCs from 12 birdshot uveitis patients (BCR-UV) and 12 healthy controls (HC) are analyzed by single-cell RNAseq. Dot plot shows two-sided proportion test analysis of NK cell clusters gated from the C4_Natural Killer cells cluster in Fig. 2A and comparing between BCR-UV and HC. Significant differences at a false discovery rate of 5% [FDR < 0.05] and log2 fold difference = [Log2(FD)], are indicated in red. The data points represent log2 fold difference, with error bars showing the estimated 95% confidence interval, based on the distribution of bootstrapped log2 fold differences. **B** Heatmap of the gene expression profile of highly expressed genes ($n = 10$ unique genes) in clusters 1, 2, 6, and 10 of NK cells as identified in *A*.

11) and a minor population was CD56$^{bright}$ and CD16$^-$ (*cluster* 3, 4; Supplementary Fig. 4B). These 12 flow cytometry clusters broadly intersected with the NK clusters detected by the scRNAseq analysis: scRNAseq clusters 1 and 3 (Fig. 2C) are represented by the flow cytometry clusters 3 and 4 (Fig. 4A), all of which are CD56$^{bright}$ population. The remaining 10 clusters in both scRNAseq and flow cytometry analyses are CD56$^{dim}$ and predominantly CD16$^+$ with the exception of cluster 2. Differential cluster abundance analysis revealed that clusters 4 and 5 were significantly reduced in the BCR-UV, while cluster 0 was significantly increased (Fig. 4B, C and Supplementary Fig. 4C). The CD56$^{bright}$ cluster 4 was further defined by high expression of CD336 and CD94 whereas the expanded CD56$^{dim}$ CD16$^+$ cluster 0 was defined by high co-expression of CD8 and CD244 and was relatively more abundant in BCR-UV blood (Fig. 4D). We further found that CD8$^+$CD244$^+$ cluster 0 expressed surface markers CD314 (NKG2D) and CD337 (NKp30), but not CD57 (Fig. 4D, E). The principal component analysis supported that cluster 0 was best distinguished by co-expression of CD8a and CD244 (Fig. 4E). Finally, we quantified the CD8a$^+$CD244$^+$ NK cells in BCR-UV patients versus healthy controls in the flow cytometry data by manual gating. This analysis revealed significantly increased frequency of CD8a$^+$CD244$^+$ NK cells in patients compared to controls (Fig. 4F).

We determined by in vitro culture that similar to the NK activation marker CD69, CD244 expression was also upregulated in purified NK cells after culturing with cytokines, IL-15 and IL-18, for two days (but not with cytokine CXCL-16, Fig. 5A), supporting that CD244 expression indicates activated NK cells. Further, we cultured PBMCs from three different healthy donors in the presence of IL-15 and CXCL-16 dilutions for 48 h. We then gated on the CD8$^+$ NK cells and checked for the expression of CD69. We observed a consistent increase of CD69 level in CD8$^+$ NK cells when treated with IL-15 but not with CXCL-16 (Supplementary Fig. 4D). Interestingly, IL-15 even in the highest dilutions showed considerable elevation of CD69 level indicating that the cytokine is highly potent to induce activation of CD8$^+$ NK cells (Supplementary Fig. 4D).

We observed that CD8$^+$ NK cells from the BCR-UV patients indeed produced higher levels of TNF and IFN-γ compared to controls (Fig. 5B),

indicating that the CD8a$^+$ CD244$^+$ cluster identified by flow cytometry may represent activated CD8$^+$ NK cells. In support of this hypothesis, we observed an overall increase in the surface expression of activating receptor NKG2D (CD314) and the cytotoxicity receptor (NCR) NKp30 (CD337)[32] in BCR-UV NK cells (Supplementary Fig. 4E) by flow cytometry analysis. Specifically, the CD8$^+$ CD244$^+$ NK cells in BCR-UV patients showed a significant elevation of NKG2D and NKp30 expression compared to that of age-matched healthy controls (Fig. 5C), supporting that this NK cell subset is activated in patients. We also reanalyzed the scRNAseq data; there are a total of 9619 NK cells (Fig. 2A, C4_Natural Killer cells) in our dataset. Of the 9619 cells, 296 express at least 1 transcript copy of *CD8A* and *CD244*, which we considered collectively as CD8$^+$ CD244$^+$ NK cells. Looking into the transcriptome of these *CD8A$^+$CD244$^+$* cells, we found significantly elevated expression of a panel of inflammatory NK genes, including *KLRK1, KLRF1, CD226, PLCG2, CSF2, TNF*, and *TBX21* in BCR-UV, compared to the healthy controls (Fig. 5D). Collectively, this data shows that CD8$^+$ CD244$^+$ NK cells are activated in patients compared to their healthy counterparts.

A recent study by McKinney et al. identified CD8$^+$ NK cells in multiple sclerosis (MS) patients that were associated with a more favorable clinical outcome[28]. We compared the hallmark genes from the CD8$^+$ NK cells identified in their study (77 genes) with the expression profile of the CD8$^+$ NK cells identified in this study but did not find significant overlapping genes (2 overlapping genes, Supplementary Fig. 4F). This analysis suggests that the CD8$^+$ CD244$^+$ NK cells found in BCR-UV patients represent a different population as compared to those reported by McKinney et al. [28]. We noted that CD8$^+$ NK cells are found in both the CD56$^{dim}$CD16$^+$ (approx. 40%) and CD56$^{bright}$CD16$^-$ (approx. 20%) NK subsets (Supplementary Fig. 4G), which may in part explain these differences.

### CD8a$^{bright}$ CD244$^{bright}$ cytotoxic NK cell frequency correlates with disease activity

Finally, we were interested to determine the dynamics of the newly identified NK cell subset in BCR-UV patients over the course of the disease. We had the opportunity to analyze samples taken prior to

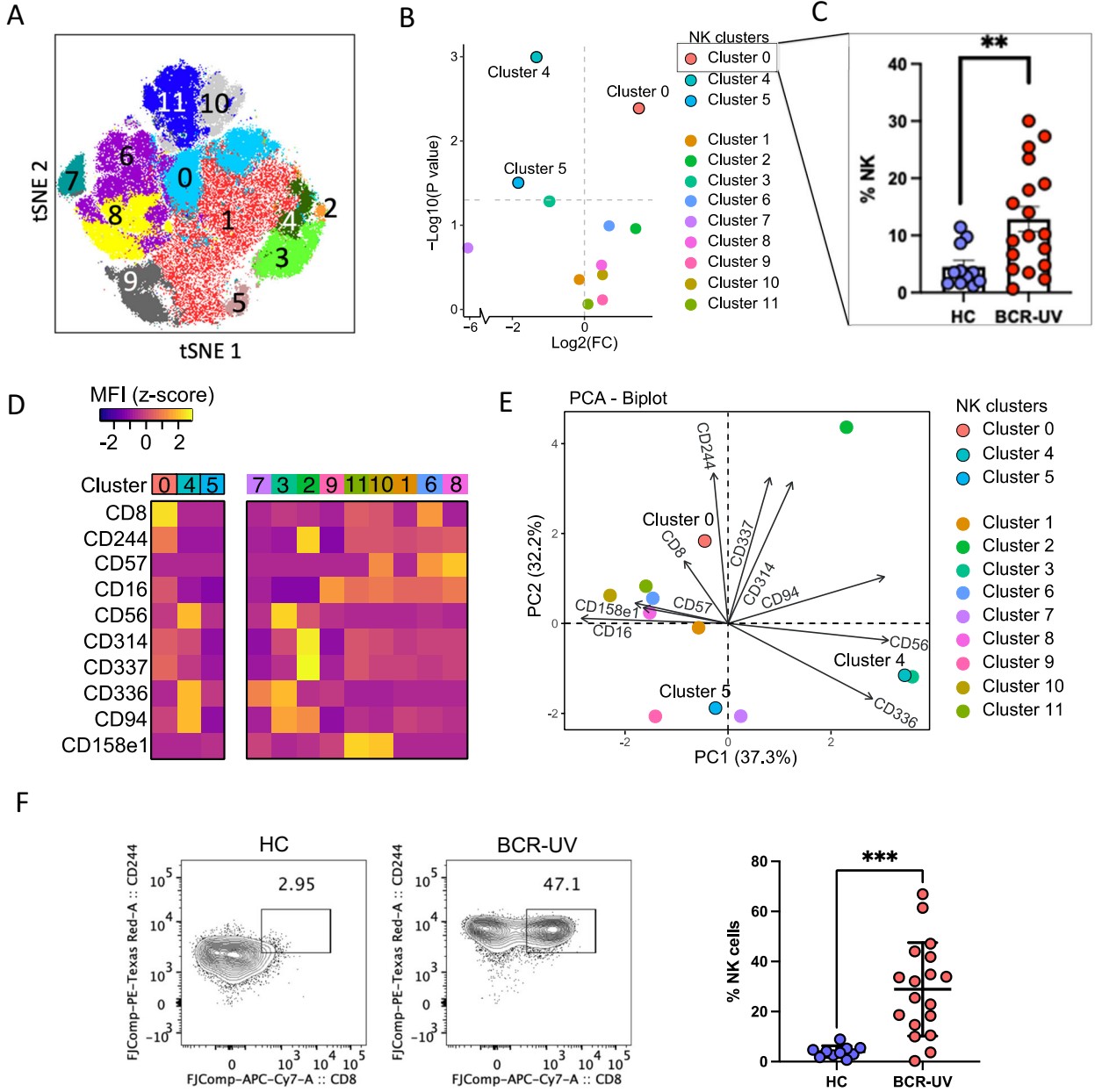

**Fig. 4 | CD244^bright CD8^bright NK cell subset is significantly expanded in peripheral blood of birdshot uveitis patients. A** t-SNE plot of flow cytometry data from Healthy control (HC) $n = 11$, birdshot uveitis (BCR-UV) $n = 18$. Clusters identified by FlowSOM analysis ($n = 12$ clusters) are shown. **B** Comparison of the frequency of NK clusters between BCR-UV and Healthy controls. The -log10 (nominal $P$ values) are from a two-sided likelihood ratio test. Significantly different clusters 0, 4, and 5 are highlighted. Source data are provided as a Source Data file. **C** Scatter plot of the frequency of cells of cluster 0 in peripheral blood in patients versus controls. HC $n = 11$, BCR-UV $n = 18$. ***$P = 0.002$. Data are presented as mean values ± SEM. Source data are provided as a Source Data file. **D** Heatmap showing the mean fluorescent intensity (MFI) of surface markers in NK clusters. Source data are provided as a Source Data file. **E** Principal component analysis (PCA) of NK clusters based on expression of surface markers. Source data are provided as a Source Data file. **F** Representative dot plots and corresponding frequency of CD8a^+CD244^+ NK cells in the peripheral blood of Healthy controls (HC, blue) and birdshot uveitis patients (BCR-UV, red). HC, $n = 10$; BCR-UV, $n = 15$. Statistical comparison is done using two-sided unpaired $t$ test. ***$P = 0.0003$. Data are presented as mean values ± SEM. Source data are provided as a Source Data file.

commencing therapy, during, and upon achieving clinical quiescence (according to the SUN criteria[33]) following treatment with systemic immunomodulatory therapy (1-year follow-up; Fig. 6A, Supplementary Fig. 5A). We assessed the expression of CD8a and CD244 in the CD56^dimCD16^+ cell population in this cohort by flow cytometry. The mean fluorescent intensity (MFI) of surface expression for both CD8a and CD244 was elevated in patients with active BCR-UV and decreased over the course of treatment (Fig. 6B and Supplementary Fig. 5B). Similarly, the CD8a and CD244 double-positive cells within CD56^dimCD16^+ NK population were significantly increased in patients with active BCR-UV and gradually decreased

with one year of treatment and normalized to the frequency observed in healthy controls ($P < 0.05$; Fig. 6C, D). While looking at the changes in CD56 bright/dim NK cells populations after treatment, we observed a moderate increase in the frequency of CD56^bright and concomitant decrease in frequency of CD56^dim population in some patients during the course of treatment (Supplementary Fig. 5C); however, these changes were not statistically significant. In conclusion, these results show that CD8a^bright CD244^bright CD56^dim cNK cells are expanded during active uveitis in BCR-UV patients but decrease upon successful systemic immunomodulatory treatment and clinical remission, compatible with the

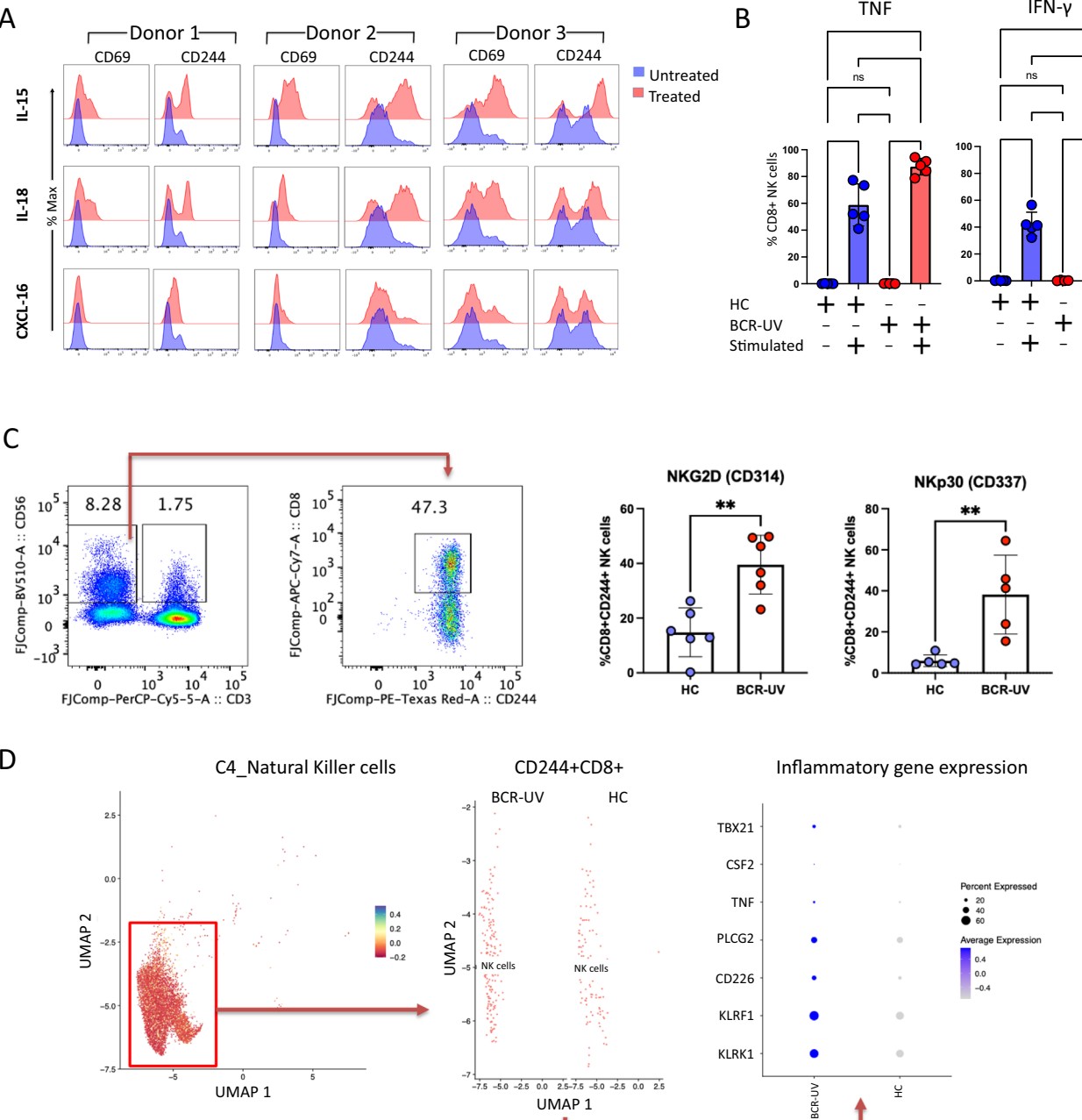

**Fig. 5 | IL-15/IL-18-mediated stimulation increased CD244 expression in NK cells and CD8 + NK cells birdshot uveitis patients show elevated activation profile.** **A** Histograms of the surface expression of CD69 and CD244, after 48 hrs stimulation of isolated peripheral blood NK cells from healthy donors. with recombinant human IL-15, IL-18 and CXCL-16 proteins. n = 3 **B** Frequencies of TNF and IFN-γ producing CD8+ NK cells, as determined by flow cytometry analysis upon stimulation with Lymphocyte Activation Cocktail (BD Biosciences). Analysis was conducted using PBMCs from BCR-UV (red) and NK cells from healthy controls (HC, blue). HC n = 5; BCR-UV n = 5. Statistical analysis was done using one-way ANOVA-Tukey's multiple comparisons test; ****P < 0.0001, ***P = 0.0003. Data are presented as mean values ± SEM. Source data are provided as a Source Data file.

**C** CD56+ CD8+CD244+ NK cell population (representative gating strategy is shown on the left) from age-matched BCR-UV patients and healthy controls (HC) are tested for the surface expression of NKG2D (CD314) and NKp30 (CD337) proteins (bar plots on the right). n = 6. **P = 0.001. Data are presented as mean values ± SEM. Source data are provided as a Source Data file. **D** *CD244+CD8A+* cells from C4_Natural Killer Cells cluster in the single-cell RNAseq analysis (subcluster shown as in Fig. 2A) were analyzed for the expression of transcripts of the indicated inflammation-associated genes in BCR-UV and HC. The double positive population expressing at least one transcript copy of *CD244* and *CD8A* were more in BCR-UV and expressed more inflammation-associated genes compared to the HC.

interpretation that CD8a^bright CD244^bright CD56^dim cells are a pro-inflammatory NK subset that are likely to be involved in the underlying disease mechanism.

## Discussion

In this study, we conducted deep molecular phenotyping of peripheral blood NK cells and identified altered changes in CD56^dim/CD56^bright

subsets in peripheral blood of BCR-UV patients. We found expansion of a CD56^dim CD8^bright CD244^bright NK cell subset in BCR-UV patients that decreased upon successful treatment with systemic immunomodulatory therapy. Our findings also corroborate previous reports on the decreased CD56^bright NK cells in BCR-UV[17] and the normalization of CD56^bright NK cell abundance upon immunosuppressive treatment of non-infectious uveitis[18]. In other MHC-I associated conditions, such as

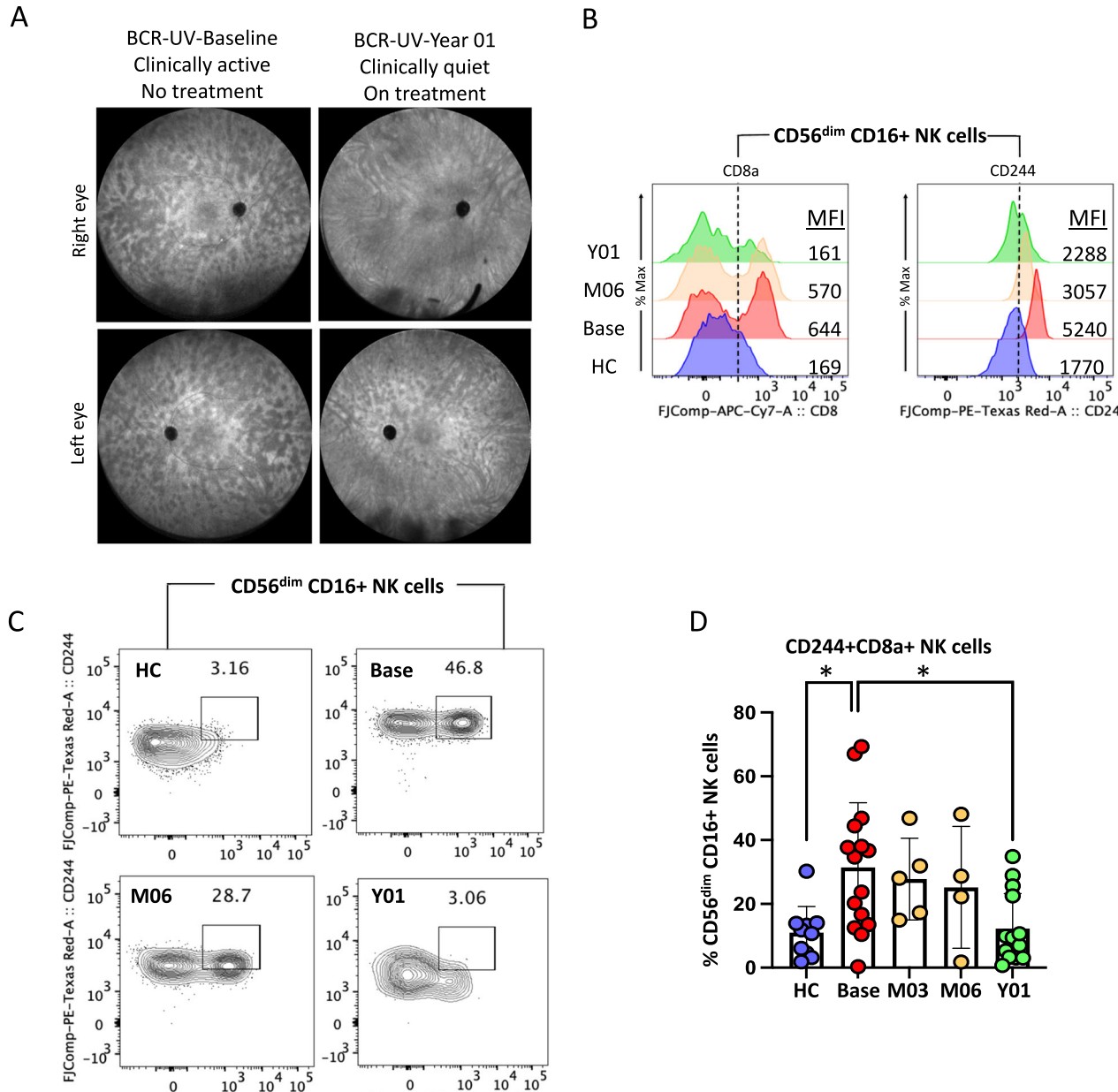

**Fig. 6 | Response to systemic immunomodulatory therapy is accompanied by normalization of circulating CD8+ CD244+ NK cells in patients. A** Representative Fluorescein angiography (FA) image the left and right eye of a birdshot uveitis (BCR-UV) patient at baseline and after 1 year of systemic immunomodulatory therapy. **B** Histograms are showing the expression of CD8a and CD244 proteins in circulating CD56dim CD16+ NK cell surface of healthy control (HC), BCR-UV patient at the baseline (Base) and on month 6 (M06) and year 1 (Y01). Values within the box indicate the mean fluorescent intensity (MFI) of CD8a and CD244 proteins. **C** Representative flow plots show staining of CD8 and CD244 proteins within

CD56dim CD16+ NK cells in the blood of healthy control (HC), BCR-UV patient at baseline with no treatment (Base), and longitudinally at follow-up visits at month 6 (M06) and year 1 (Y01) after treatments. **D** Frequencies of CD244 and CD8 double positive cells within CD56dim CD16+ NK population in age-matched healthy control (HC) and in BCR-UV at baseline (Base) and treatment follow up visits. HC, $n = 10$; Baseline, $n = 15$; Month 03, $n = 4$; Month 06, $n = 4$ and Year 01, $n = 15$. Statistical comparison is done using one-way ANOVA- Tukey's multiple comparisons test. *$P = 0.01$. Data are presented as mean values ± SEM. Source data are provided as a Source Data file.

HLA-B51-associated Behcet's disease (BD) and HLA-B27-associated ankylosing spondylitis (AS), CD56dim/CD56bright changes have been reported. In BD, total NK cells are increased in frequency in blood[34,35] and produce increased IFN-γ and TNF[36,37], which we also demonstrate in BCR-UV. In AS, the number of CD56dim CD16+ subset of NK cells in the peripheral blood is increased[38–40], which is in line with our finding of elevated CD56dim CD16+ NK cells in BCR-UV. Here, we add to these previous observations that the decline in CD56bright regulatory NK cells in BCR-UV is accompanied by a concomitant expansion of CD56dim CD16+ NK cells, that also express the alpha-chain of the CD8 co-

receptor for HLA class I[41,42], advancing the understanding of NK cell dynamics in ocular inflammatory disease. The subset of CD8+ NK cells[28] make up approximately half of the blood NK cells[43]. Note that in contrast to CD8+ T cells, which express the alpha (α) and beta (β) chain of CD8, human CD8+ NK cells only express two α-chains of CD8 and mouse NK cells do not express CD8 at all[41,42]. In T cells, CD8 helps the recognition of the specific MHC-peptide (pMHC) complex by the T cell antigen receptor (TCR). pMHC-I molecules are major ligands for both forms of CD8 and the binding affinity of CD8αα/pMHC is comparable to that of CD8αβ[44,45]. Similar to CD8αβ a co-receptor for TCR, CD8αα

also functions as a co-receptor for KIRs in NK cells and enhances pHLA-I binding to, for example KIR3DL1, in an expression-dependent manner[46]. Moreover, CD8αα also binds with MHC class I molecules expressed on NK cells in the NK–NK synapse that leads to the rescue of CD8+ NK cells from secretion-induced apoptosis- a mechanism that involves CD8-induced influx of calcium from the extracellular sources[47].

CD8+ NK cells are considered to be functionally distinct and produce more IFN-γ and TNF compared to CD8− NK cells[48]. Accordingly, our data show that the CD8+ NK cells-enriched population in patients secretes more IFN-γ and TNF compared to healthy controls. In addition, the activation marker CD69 was induced upon treatment with IL-15 or IL-18 (Fig. 5A). This is significant because cross-linking of CD8 on NK cells induces increased expression of the activation marker CD69 at the cell surface[47]. Of interest, elevated frequency of peripheral blood CD8+ NK cells with homing marker CXCR3 is associated with an increased risk for Type 1 diabetes, a T-cell mediated autoimmune condition[49]. The most expanded CD8+ NK subpopulation identified by scRNAseq (cluster 10) in BCR-UV patients was also characterized by high CXCR3 expression (Supplementary Fig. 3D). Whether this CD8+ NK subset (i.e., cluster 10) may directly contribute to eye inflammation in BCR-UV, remains to be determined. Alternatively, the CD8+ NK skewing may be a reflection of diminished regulatory capacity in the NK cell compartment. NK cells are required to suppress CD8+ T cell autoimmunity[50] which is attributed to the negative immunoregulation of activated T cells by CD56bright NK cells[51]. As shown in this study, CD56bright NK cells are diminished in circulation of NIU patients, including BCR-UV patients. This makes it tempting to speculate that proinflammatory skewing of the NK cell population diminishes control of autoreactive CD8+ T cell immunity directed towards the eye in BCR-UV. The transcriptomic heterogeneity that we have observed in NK cell subsets suggests their functional diversity. Differential presence/absence of NK cell subsets in the BCR-UV population is also indicative of possible pathogenic responses. For example, MYOM2 and IGFBP7 (as well as other genes identified in our study, such as SH2D1B) are top marker genes for NK cell subpopulations that correlate with disease severity in COVID-19[52]. Furthermore, MYOM2 has also been identified in NK cell subset in single-cell transcriptomics in tuberculosis, and Alzheimer disease[53,54]. The existing literature suggests that SH2D1A (SLAM-associated protein or SAP) and SH2D1B (EAT2) are SH2-domain-containing signaling molecules that function downstream of CD244 signaling and promote NK cells activation[31,55]. SH2D2A (TSAd; highly expressed in NK cell cluster 10, Fig. 3B) is also preferentially expressed in activated T cells and NK cells[56,57]. This protein has been found to be involved in multiple signaling pathways, including those of the T-cell receptor[57–59], however, it's role downstream of CD244 has not been explored yet.

Detailed immunoprofiling by scRNAseq and flow cytometry revealed that expanded CD8+ NK cells in BCR-UV co-express high levels of other functional receptors, including CD244, which we demonstrate is a marker for activated NK cells. CD244 [or SLAMF protein 2B4] is an immunoregulatory surface receptor expressed by NK cells and T cells[60,61]. CD244 was first discovered in NK cells and CD8+ T cells as a stimulatory cell surface receptor that mediated non-MHC-restricted killing by the lymphocytes[62–64]. However, later it was suggested that CD244-deficient murine NK cells were more cytotoxic than wild-type NK cells[65]. A model proposed based on the relevant discoveries suggests that the interaction of CD244 on NK cells with its ligand CD48 expressed by other cells leads to the signal transmission either through activation or inhibitory intracellular signaling proteins[66]. For instance, binding of CD244 intracellular motifs to SAP upregulates the viability and cytotoxic effects of NK cells and CD8+ T cells, but its binding to EAT-2 or EAT-2-related transducer (ERT) may transmit inhibitory signals[66]. Since mature human NK cells abundantly express SAP[67], the role of CD244 in human NK cells appears to have activating functions.

However, mature mouse NK cells express SAP, EAT-2, and ERT[68] leading the receptor to contribute to either stimulation or inhibition NK cell activation[66]. We found a clear association of BCR-UV with CD244 intracellular adaptors EAT-2 and TSAd encoded by SH2D1B and SH2D2A, respectively, which also suggests a dual nature of CD244 signaling in BCR-UV. However, concomitant expression of TNFRSF18 (Tumor Necrosis Factor Receptor Superfamily, Member 18) and ISG15 (Interferon-Stimulated Protein, 15 kDa) levels in the NK subset supports the 'inflammatory' phenotype of this NK subset in BCR-UV patients.

We showed that the CD244bright CD8+ NK cells are associated with disease activity during longitudinal monitoring of patients treated with systemic immunomodulatory therapy. The CD8+ NK cell frequency has previously been shown to correlate with clinical parameters in several conditions, including HIV-1 and multiple sclerosis (MS)[28,48]. More specifically, in MS, the CD8+ NK cell frequency was predictive of the relapse rate in a longitudinal cohort[28]. It would be interesting to use flow cytometry analysis of this cell subset to see if its abundance can be used to predict treatment outcome or clinical course in advance (measured at diagnosis).

In conclusion, using complementary immunophenotyping platforms, we identified an expanded and activated CD8abright CD244bright population of circulating NK cells in BCR-UV whose abundance reflects inflammatory disease activity. Better understanding of the molecular underpinnings of BCR-UV and its relation to clinical outcome may pave the way towards the implementation of more effective personalized therapeutic approaches in ocular inflammatory diseases.

## Methods
### Patients
This study was conducted in compliance with the Declaration of Helsinki and ethical principles regarding human experimentation. All samples were obtained under a National Institutes of Health (NIH) Institutional Review Board (IRB) approved protocol (Uveitis/Intraocular Inflammatory Disease Biobank, iBank; NCT02656381). Informed consent was obtained from all enrolled participants or from participant's parents/legally authorized representative in case of minors. We recruited 139 patients (Supplementary Data 1) with non-infectious uveitis (NIU), including 18 patients with birdshot chorioretinopathy uveitis (BCR-UV) at the National Eye institute (NEI) outpatient clinic. In total, 80 healthy donors- majority (iBank IDs) of whom were screened for no personal or family history of autoimmune diseases, were recruited and served as unaffected healthy controls. NIU was classified and graded in accordance with the SUN classification[33]. All patients with BCR-UV were HLA-A29-positive confirmed by HLA typing. A retrospective review of patient charts, fluorescein angiography (FA), indocyanine green angiography (ICGA), and electroretinography (ERG) was performed in order to determine disease activity, clinical course and response to systemic immunomodulatory treatment. Patient demographics are summarized in Supplementary Data 1. Sexes of all study participants were determined based on self-reports and have been reported with consent. There is an almost equal distribution of participants of both sexes in this study. There is no previous report of NK cells discriminating among the genders. Therefore, sex and/or gender were not considered in the study design.

### Blood sample processing
Blood samples from patients and healthy controls were collected through venipuncture and all the samples were processed within 4 h of blood collection. Three mL of fresh whole-blood samples were directly used for flow cytometry staining and acquisition. PBMCs were purified by standardized density gradient isolation (Ficoll-Paque) and stored in liquid nitrogen until further use. NK cells were isolated from PBMCs using human NK cell isolation kit (Miltenyi Biotec, Cat# 130-092-657).

## Flow cytometry

Three mL of whole blood was incubated with 30 mL 1x RBC lysis buffer (BioLegend #420391) at room temperature for 15 minutes, centrifuged at 400x g for 5 minutes and resuspended in a 30 ml FACS buffer (FB: 1x PBS w/o calcium and magnesium chloride + 2 % FBS + 2 mM EDTA + 0.01 % NaN₃). Cells were counted and in total 3×10⁶ cells were stained for 45 minutes following resuspension in 100 uL FACS buffer with Fc block (Human TruStain FcX, Biolegend # 422302) with the following antibodies (Supplementary Data 2): Alexa Fluor 488 anti-human CD3 Antibody (clone OKT3, Cat# 317310, Biolegend, dil 1:300), Alexa Fluor 700 CD16 (clone 3G8, Cat# 302026, Biolegend, dil 1:100), APC anti-human CD57 Antibody (clone HNK-1, Cat# 359610, Biolegend, dil 1:200), APC-Fire 750 CD14 (clone 63D3, Cat# 367120, Biolegend, dil 1:200), APC-Fire 750 CD8 (clone SK1, Cat# 344746, Biolegend, dil 1:300), Biotin anti-human CD19 Antibody (clone HIB19, Cat# 302204, Biolegend, dil 1:300), Biotin anti-human CD20 Antibody (clone 2H7, Cat# 302350, Biolegend, dil 1:300), Biotin anti-human CD3 Antibody (clone SK7, Cat# 344820, Biolegend, dil 1:300), Biotin anti-human CD56 Antibody (clone HCD56, Cat# 318320, Biolegend, dil 1:200), BV 421 anti-human CD158e1 Antibody (clone DX9, Cat# 312714, Biolegend, dil 1:300), BV 421 anti-human CD19 Antibody (clone HIB19, Cat# 302234, Biolegend, dil 1:300), BV 510 anti-human CD20 Antibody (clone 2H7, Cat# 302340, Biolegend, dil 1:300), BV 510 anti-human CD56 Antibody (clone HCD56, Cat# 318340, Biolegend, dil 1:200), BV 650 anti-human CD314 Antibody (clone 1D11, Cat# 563408, BD, dil 1:50), BV 650 anti-human HLA-DR Antibody (clone L243, Cat# 307650, Biolegend, dil 1:100), FITC anti-human CD4 antibody (clone SK3, Cat# 344604, Biolegend, dil 1:300), FITC anti-human CD94 antibody (clone DX22, Cat# 305504, Biolegend, dil 1:300), PE anti-human CD336 Antibody (clone P44-8, Cat# 325108, Biolegend, dil 1:50), PE/Cy7 anti-human CD337 Antibody (clone AF29-4D12, Cat# 25-3379-42, Thermo Fisher, dil 1:50), PE/Dazzle 594 anti-human CD244 Antibody (clone C1.7, Cat# 329522, Biolegend, dil 1:300), PerCP/Cy5.5 anti-human CD3 Antibody (clone SK7, Cat# 344808, Biolegend, dil 1:300). BV 605 Streptavidin (1:100) and PE/Cy7 Streptavidin (1:500) were used for secondary staining. The PE/Cy7 Lin cocktail in monocyte/DC panel includes anti-human CD3, CD19, CD20 and CD56 Antibodies. Live/dead staining was carried out using Fixable Viability Dye eFluor 455UV concomitantly with surface staining. Cells were fixed with Fixation Buffer (Biolegend, Cat #420801) and resuspended in 300 uL FACS buffer from where a set volume (200 uL) of cells were acquired on a BD LSR Fortessa within 1-3 days of staining. The dates of flow-staining and acquisition are indicated in Supplementary Data 1.

Data were analyzed using FlowJo v10. Flow cytometry gating was done as- Lymphocytes (SSC-A vs FSC-A plot)/Single Cells (FSC-H vs FSC-A plot)/Single Cells (SSC-H vs SSC-A plot)/Live (Aqua L/D negative). CD20 negative (CD20 vs FSC-A plot) and CD56 positive (CD56 vs CD3 plot) cells are gated as NK cells (Supplementary Fig. 1E). Data shown % Live NK cells are the frequency of CD56+ cells within the Live gated cells.

For ex vivo restimulation, NK cells from a healthy donor were cultured with recombinant human IL-15 and IL-18 using two-folds serial dilutions of ED₅₀ concentrations (1:1). Cell-surface expression of CD69, CD244, and CD8 was measured using a BD LSR Fortessa and analyzed by FlowJo V.10.

## Intracellular cytokine staining

For intracellular cytokine staining, cryopreserved PBMCs were thawed into warm RPMI/10% FBS, washed once in cold PBS and divided cells in two equal proportions. Protein Transport Inhibitor Cocktail (eBioscience, Cat#00-4980-03, dil 1:500) was added to each proportion. Cells were incubated with cell stimulation cocktail (eBioscience, Cat# 00-4970-03, dil 1:500) for 4 hours at 37 degrees or kept under similar conditions without the stimulation cocktail. Cells were washed with cold PBS and resuspended in 100 uL FACS buffer with Fc block (Human TruStain FcX, Biolegend, Cat# 422302) followed by surface markers staining for 45 min. Cells were then washed once in cold PBS and incubated in fix/perm buffer (eBioscience FOXP3/Transcription Factor Staining Buffer, Cat #00-5523-00) for 20 min and then incubation with the following intracellular antibodies for another 30 min: BV510 anti-human Granzyme B (clone GB11), APC/Cyanine7 anti-human TNF (clone MAb11), PE/Dazzle 594 anti-human IFN-γ (clone 4 S.B3) and PE anti-human Perforin (clone B-D48).

## PBMC single-cell RNA sequencing

PBMCs from a total of 24 BCR-UV patients and healthy controls (Supplementary Data 3) were thawed quickly at 37 °C and resuspended in RPMI media supplemented with 10% FBS. Approximately, 1,000-1,200 viable cells per microliter were loaded for capture onto the Chromium System using the v3 single-cell reagent kit (10X Genomics, Chromium Next GEM Single Cell 3′ GEM, Library & Gel Bead Kit v3.1, 16 rxns PN-1000121). Following capture and lysis, scRNA-Seq libraries were prepared from ~16k cells using a 10X Genomics Chromium device and Chromium Single Cell Reagent Kit (10X Genomics, Chromium Next GEM Single Cell 3′ Library Kit v3.1 16 rxns PN-1000157) according to manufacturer's protocol. GEX (transcriptome) libraries were sequenced on a NovaSeq 6000 DNA sequencer (Illumina, Inc.) using V1.5 chemistry, generating ~160 M GEX reads per sample and ~40 M ADT reads per sample. The raw data was processed using RTA 3.4.4. Sequencing information is shown in Supplementary Data 4.

## Preprocessing of scRNAseq data

The raw fastq data were processed using *cellranger* v6.0.0[69] with genome assembly GRCh38 (hg38), 3′ and 5′ assay chemistry SC3Pv3, and an expected cell count of 10,000 per sample (Supplementary Data 5). Further analysis of the single-cell data was done using the *Seurat* v4.0.5[70] package in the R v4.1.0 environment[71]. Cells with fewer than 300 genes and number of transcripts counts less than 1,000 and more than 12,000, more than 8% mitochondrial and 40% ribosomal fraction were excluded from the dataset. Doublets were detected and removed using the *doubletFinder v3.0* package[72] in R, set with an expected doublet level of 7%. Data were normalized using the "Log-Normalize" method with the scaling factor set at 10,000. The variables, sample batch, percent mitochondria, percent ribosomes, transcript counts and gene counts, were regressed out using the ScaleData function.

## Dimension reduction, clustering and visualization

Cells were clustered using Principal Component Analysis (PCA) using the RunPCA function. The first 20 PCs identified (by Elbow method) were used in the 'FindNeighbors' (based on k-nearest neighbor (KNN) graphs) and 'FindCluster' (Louvain algorithm) functions in Seurat. RunTSNE and RunUMAP functions were used with "pca" as the reduction method, to visualize the data. The *FindAllMarkers* function was used to identify cluster-specific markers. Cell type annotation was generated manually combining automatic annotation results from ScType[73], with "Immune System" set as the tissue type, and *SCSA*[74], with the whole database as a reference. Outcomes from ScType and SCSA were used to curate cluster annotation and identify NK cells by plotting the expression of lineage-specific genes.

## FlowSOM analysis

Live NK cells (Aqua L/D-CD3-CD20-CD56+) were gated from 18 BCR-UV patients sampled at baseline (i.e., at disease onset or relapse) and 10 healthy controls. NK cell fraction (range 5,011 to 42,358 NK cells) was down-sampled to 5,000 NK cells per donor using FlowJo Down-sampleV3 plugin, concatenated and subjected to t-distributed stochastic neighbor embedding (*t-SNE*, iteration 1000, perplexity 30). The FlowSOM plugin in FlowJo was used to cluster cells into 12 meta-

clusters (following 12 clusters identified in scRNAseq). The frequencies of the 12 NK cell meta-clusters are added as Supplementary Data 6.

## Statistical analysis

Quantification and data analysis of experiments are expressed either as median with ranges (box and whiskers plot) or as mean ± standard deviation (bar and dot plots). P values were calculated using analysis of variance (ANOVA- Tukey's multiple comparison test) or two-tailed Student's t-test for pairwise comparisons or non-parametric Mann-Whitney U test or Kruskal-Wallis test with Dunn's multiple comparison and were calculated using Graphpad Prism v.9. Differences in proportions of scRNAseq data between the groups were assessed using the "scProportionTest" package[75], with number of permutations set at 10,000. Qualitative experiments were repeated independently to confirm accuracy.

## Systemic therapy of patients

Details of individual patient's treatment category, clinical and angiographic activity as well as the systemic immunomodulatory therapy are included in Supplementary Data 7.

## Reporting summary

Further information on research design is available in the Nature Portfolio Reporting Summary linked to this article.

## Data availability

Source data are provided as a Source Data file and mentioned in all relevant figure legends. All raw data and workflow are available as an open-source resource, with documentation. The primary data used in this research is deposited and available for the public, without any restrictions, at the NCBI SRA repository under the accession PRJNA855114. All other data are available in the article and its Supplementary files or from the corresponding author upon request. Source data are provided with this paper.

## Code availability

Data analysis was done using custom workflow scripts, written using R and BASH programming languages. The workflow components include cellranger, Seurat, SCSA, SCTYPE, Plotly, ggplot and other dependent packages. This work utilized the computational resources of the NIH HPC Biowulf cluster. Entire code, example demo data and the detailed documentation are available for public use at the github site https://github.com/PulakNath/bcr-uveitis. In addition, a permanent reference to the version of the code used in this study has been generated by linking the github repository to Zenodo[76].

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

## Acknowledgements

The authors acknowledge the NIAMS Flow Core, NEI Flow Core and NHLBI flow core (special thanks to Dr. Pradeep Dagur for flow sorting and generating data in response to reviewers' comments) for help with this project. We also thank Dr. Lotta Utriainen for setting up the initial flow panels. The work is funded by the NEI intramural research program (project number 1ZIAEY000556-04) and made possible by the Lasker Clinical Research Grant to H.N.S., and the Vision Grant from Prevention of Blindness Society of Metropolitan Washington (years 2020 and 2021) to P.R.N.

## Author contributions

P.R.N. did the study conception and design, collected data, analyzed and interpreted the results, and wrote the manuscript. M.M., J.W.L., M.Y., A.K., H.N. and S.K. did the data generation and analysis. B.S. and A.Y. generated the single-cell RNAseq data. V.N. and K.D.K. did in silico analysis of the data. S.K. and H.N.S. provided the patient care and provided the patient samples for the study. R.R.C. did the data interpretation and reviewed the manuscript. J.J.W.K. and H.N.S. jointly supervised the work and did the analysis and interpretation of results and wrote the manuscript.

## Competing interests

The authors declare competing interests.
