## [Peer Review File · Nature Communications]

Single-cell profiling identifies a CD8^{bright} CD244^{bright} Natural Killer cell subset that reflects disease activity in HLA-A29-positive birdshot chorioretinopathyREVIEWER COMMENTS

Reviewer #1 (Remarks to the Author):

The authors analyzed circulating major lymphocyte subsets in 18 BCR-UV using flow cytometry and found that frequency of NK cells in PBMC was significantly higher in BCR-UV compared to that of healthy controls. Among the NK cells of BCR-UV, CD56dim CD16+ NK cells were higher, whereas CD56bright CD16- NK cells were lower than those of controls. Using single cell RNA sequence analysis on 12 BCR-UV and 12 healthy controls, the authors classified NK cells into 10 NK subclusters and found that the cluster 1 was less frequent and clusters 2, 6, 10 were more frequent in BCR-UV. As CD244 and CD8A showed relatively higher expression on the clusters 2, 6, 10, the authors analyzed these molecules on NK cells using flow cytometry and confirmed that CD8 and CD244-expressing NK cells are more frequent in BCR-UV. Furthermore, the expression of CD8 and CD244 were decreased after immunomodulatory therapy.

This study is noteworthy in that advanced techniques such as single cell RNA sequence and high-dimension flow cytometry were used to understand immune profiles of peripheral blood of BCR-UV. The finding that CD56dim NK cells are enriched in PBMC of BCR-UV is interesting. CD8+CD244+ NK cells were more frequent in active BCR-UV and the NK subset decreased after immunomodulatory treatment are intriguing. However, there are many concerns in the concept of NK cell subsets and the data interpretation which require more improvements.

Major comments

1. Although BCR-UV is strongly associated with HLA-A*29, HLA-A*29 is not recognized by NK cell receptors, which is distinct from other uveitis associated with HLA class I. CD244 and the ligand, CD48, are not involved in HLA recognition. Although CD8 is a co-receptor for TCR, the recognition is not HLA allele-specific. Therefore, the findings in this study are not much associated with mechanisms of "MHC-I -opathies". The authors should make this point clearer in the manuscript.
2. The authors often used NK1, NK2 as CD56dimCD16, CD56brightCD16-, respectively, however, it is confusing. NK1 and NK2 were previously proposed as Th1 and Th2-like NK cells in some researchers, and recently, one group named "NK1 and NK2" as "CD56dim and

CD56bright NK cells”, respectively, however, these are not widely accepted. The reviewer would suggest not to use NK1/NK2. The authors should refer right references if the authors continue to use the terminology, since the reference referred dose not describe anything about NK1/NK2.

3. NK8 is also not well recognized as CD8+NK cells in this field and the function in NK cells is not well established. In page 14, line 5 from the bottom, “NK8+ cells are considered to be functionally distinct and produce more IFN-gamma and TNF-alpha compared to CD8– NK cells” is an observation in HIV patients. More discussion with previous studies on CD8+NK cells are necessary. Which data showed that the NK8+-enriched population in patients secretes more IFN-gamma and TNF-alpha compared to healthy controls as described in p14, line 4 from the bottom? The authors showed that CD69 was induced upon treatment with IL-15 or IL-18 on bulk NK cells (Fig. 3F), but not show that for CD8+NK cells.

4. CD244 is known to hold both activating and inhibitory dual functions depending on conditions, such as expression levels of CD244 and CD48 and higher expression of CD244 could result in inhibitory signal. It is not clear whether CD244+CD8+NK cells in active BCR-UV patient are really activated and involved in the inflammation in BCR-UV. In vitro stimulation of bulk PBMC using PMA/Ionomycin, IL-15, 18 are not enough to show that the subset in the patients is activated. As NKG2D and NKp30 are commonly expressed on NK cells, these expressions do not mean they are activated. The authors should show expression of some activation markers or activating functions of CD244+CD8+NK cells.

5. The authors showed changes in frequency of CD8+CD244+NK cells by the treatment course. How about the changes in CD56bright/dim NK cells populations after treatment?

6. The authors mentioned that flow cytometry was conducted 1-3 days after staining. However, fluorescent signal is getting weaker by time. It is not very reliable if data is acquired after more than one day even though they are fixed. The authors should show the date of flow cytometry analysis for each sample. Were samples from patients and healthy controls analyzed together?

7. The authors showed that NK cells of BCR-UV patients produced significantly higher proinflammatory cytokines and suggested that it was due to altered NK1/NK2 balance (reduced CD56bright NK cells)(page 9, line 10). However, it is not correct as CD56bright NK cells can produce higher amount of proinflammatory cytokines. The authors should show the capability of proinflammatory cytokine production with CD56 expression to

demonstrate if ratio of CD56brigh/dim affected the differences in responsiveness between BCR-UV and healthy controls.

8. In P9, the 2nd paragraph, the authors concluded that “these results indicate expansion of activated CD8+ NK1 subpopulation characterized by high levels of CD244-signaling molecules in the blood of patients with BCR-UV”. However, frequencies of CD8+, CD244+NK cells among the clusters 2,6,10 were only around 20% in figure 1D. It is not convincing to say that “the expanded NK1 cluster 0 was defined by high co-expression of CD8 and CD244, in line with our scRNA-seq data” (page9 bottom line).

9. The frequencies of each NK cluster classified by flow cytometry should be shown. Where are the data indicating “Differential cluster abundance analysis revealed that clusters 4 and 5 were significantly reduced in the BCR-UV” in page 9, line 4 from the bottom?

10. In Fig 3F, upregulation of CD244 and CD8a by IL15 or IL-18 seems not be concentration dependent. Furthermore, it is generally known that stimulation via single signal is not enough to activate NK cells except for via CD16. How many donors were analyzed?

11. In Fig.3G, it is critical to analyze patients and controls together to compare the CD244 and CD8a expression. Please clarify the date of analysis.

12. The details about the systemic immunomodulatory therapy conducted on BCR-UV patients should be described.

13. The authors suggested that clusters 1,6,10 uniquely expressed MYOM2, IGFBP7, LINC00996, respectively. Are these gene expression previously reported in NK cells anywhere?

14. The abstract contains several issues. CD8A is not the HLA class I restricted antigen. The authors do not show evidence or references that IGFBP7, MYOM2, and LINC00996 are high cytotoxic signatures.

15. The authors should show evidence or references that SH2D2A is a signaling molecule downstream of CD244.

Minor comments

1. In introduction, it is not correct that CD56dim NK cells produce greater amounts of pro-inflammatory cytokines (P3, line 21). It is well known that CD56bright produce more pro-inflammatory cytokines depending on stimulations.

2. Figure 1A should show a real FSC/SSC facs plot instead of the schema. Furthermore, the leukocytes with highest SSC/FSC should be granulocytes. Macrophages are not present in peripheral blood.
3. What does WDS mean in figure 1C?
4. Statistical methods should be described in figure 1F.
5. In page 6, line 8 from the bottom, why KLRF1 is not included as the most upregulated genes in BCR-UV?
6. Please highlight TNFRSF18 (also known as GITR) in the respective figure (P9, line 13). It is hard to find the gene in the figures.
7. The authors suggested that the cluster 1 classified by scRNA seq had features identical to CD56brightCD16- (p9, line 2), however, the cluster expressed CD16 in Fig 2D.
8. In page 9, line 5 from the bottom, "The remaining 10 clusters in both scRNAseq and flow cytometry analyses are CD56dim and CD16+ populations." is not correct. The flow cytometry cluster 2 lacks CD16.
9. In p14, line 9 from the bottom, "CD56brightCD16+ NK subsets" should be CD56brightCD16- NK subsets.
10. P17, line 1, "healthy controls are collected" should be "were collected".
11. Why both RBC lysis buffer (BioLegend #420391) and ACK lysis buffer (Lonza #BP10-548E) were used?
12. "Pe/Cy7" should be "PE/Cy7".
13. "LSRFortessa" should be LSR Fortessa (p18, line 6).
14. Is it correct that the Fc block was conducted in 100% FBS? (P18, line 14) If so, why?
15. Reference 19 lacks the page number.
16. Fig S1A staining panels should divide into 2 panels, lymphocytes and monocyte/DC panels. CD1c is lacking in the list.
17. Statistical methods and P values should be shown in Fig 3H, Fig S1F and S5.
18. Plot for SH2D1B is duplicated in Fig S3B.
19. The differences between active and quiet stage are not clear in the fundus photography shown in Fig.S5A.

Reviewer #2 (Remarks to the Author):

In this manuscript, the authors performed a transcriptome and FACS analysis on PBMCs from non-infectious uveitis patients and healthy controls (HCs), in order to identify altered NK cell subsets. In the transcriptome analysis, they describe 4 subsets of NK cells (out of 12) with significantly different proportions between birdshot chorioretinopathy patients (a rare form of uveitis patients) and HCs, and one of these may correspond with previously described NK8+ cells, as claimed by the authors.

By FACS analysis, the authors describe an accumulation of CD8+CD244+CD16+ NK cell subset in peripheral blood of birdshot chorioretinopathy patients.

The data are of interest, novel and the experiments are well performed. Although the disease is a rare condition, the data are of interest in the field of autoimmune diseases. Fundamentally, the findings add to the complexity of NK cell subpopulations and illustrate the importance of a better phenotyping of NK cells and cytotoxic cells in general. Furthermore, all illustrations, including the supplementary data, are well presented. The inclusion of age matched controls and the presentation of part of the data with strictly matched controls is appreciated.

A major concern relates to the actual function of the NK8+ cells in uveitis(the authors claim a proinflammatory function in the pathogenesis).

Other comments are minor but need to be addressed.

NK8+ cells

Considering the CD8+ NK cell cluster 0 (as identified by FACS), the authors claim that cluster 10 in the transcriptome analysis corresponds with the NK8+ cells described by others. However, from the heat map shown in figure S3C, there is – to my opinion - no cluster with an increased CD8a expression. Therefore, it will be important that the authors verify whether uveitis patients have an NK8+ subpopulation which correspond with the NK8+ cells described in for example multiple sclerosis patients, described by McKinney et al., Nat Commun 2021, or other available data.

Furthermore, the manuscript of McKinney et al. presented evidence for an autoregulatory role of NK8+ cells, by inhibiting activity of potentially pathogenic CD4 T cells, while in the

discussion the authors claim that the NK8+ cells found in uveitis patients are pro-inflammatory. It will be important to provide more in vitro evidence for this statement.

Minor comments:

Introduction: besides stating that different NK cell clusters have been described in blood of healthy controls, it should also be pointed out that there are tissue-specific NK cells (ILC1 cells) with a phenotype specificity.

Results: scRNA-seq: It is better to write 12 patients and 12 HC, instead of 24 patients and HC.

Legend Figure 1: Specify the P values show in panel E

Legend Figure S1:

Panel A. Specify the lineage (light green).

Panel F, phenotyping NK2, include CD16- (also in de figure itself).

Legend Figure 2 and S2:

A total of 306,425 cells: please include "PBMC" in stead of cells

Legend Figure S4:

Panel C & D: Red dotted line captures the region where NK cells are present in BCR-UV, and "relatively" absent in HC.

Figure 3:

Beside plotting cluster 0, as significantly different between HC and Uveitis patients (panel C), it would be good to also plot clusters 4 and 5 (or include FACS data of all clusters in a suppl figure).

Panel F: regarding the in vitro stimulation of NK cells, was this not with purified NK cells (or total PBMCs and gated on NK), and with cells obtained from HCs? This should better be explained in the legend and corresponding results section.

Figure 4:

Panel B: it is mentioned that MFIs histograms are plotted from patients at base, Month 6 and Year1, together with HCs, but HCs are missing in this panel.

Reviewer #3 (Remarks to the Author):

This is an interesting manuscript which presents intriguing data on a possible role for a rare subset of NK cells in the pathology of uveitis; most specifically, the rare birdshot chorioretinopathy.

I have a major issue with the use of the term NK1 vs NK2 throughout the paper. These subsets are not determined by the levels of expression of CD56 and CD16 as implied throughout and the reference #16 is not correct to justify this use of terminology (line 72). Cytokine secretion profile and CD95 expression are the relevant criteria for NK1 vs NK2. CD56⁺⁺/CD16⁻ are cytokine secreting, poorly cytotoxic NK cells which are considered to be immature. For clarity, I would edit line 78 to:

"This includes CD56^{dim} CD16⁺ NK subsets which may be "inflammatory", identified by secretion of high levels of cytokine and interferon response genes.

NK1 and NK2 cells were shown by Deniz et al (2002) to have similar cytolytic function but to be separated on the basis of the degree of Ifn- γ secretion. Both subsets expressed CD16.

The techniques used are highly appropriate for the study and are well described. However, the choice of statistical tests is not discussed nor justified and it is difficult to believe that some of the findings reach the degrees of statistical significance which are claimed. For example, Fig 1C presents "percent live" NK cells but it is not stated whether these data are means values with SDs or SEMs or medians with ranges etc etc. The red bar showing the uveitis patient data overlaps more than 50% of the HD population and is highly skewed by the three outlier data points. This does not appear to be a Normal distribution yet an unpaired Student t test has been used to assess statistical significance which is wrong.

There are similar issues with the data presented in Fig 3H. Whilst the populations of CD244 expressing NK cells are certainly different between the BCR-UV subjects in the "lo" versus the "mid" and "hi" groups; it is important to know how these were assessed for statistical significance since both the "mid" and "hi" groups are bivariate distributions and neither is a single normal distribution and cannot be tested as such with a parametric test. It looks as

though the "mid" and "hi" subjects include two distinct clusters, the lower of which are not significantly different than the "lo" group whereas the upper cluster represents a different type of patient.

The Methods section needs a detailed explanation and justification of the statistical tests used.

On a more prosaic point, it is unclear what parent population has been used for the "% live" determination. Are these data showing the percentage of live NK cells within the live lymphocytes or total PBMC?

It is also important to know whether these differences between the percentages of NK subsets in patients versus HD are reflected in the absolute numbers of NK cells per ml of peripheral blood.

The data showing changes in CD244+/CD8+ NK cell percentages during treatment are very interesting but too limited to justify the claim in line 336 that this cell population "correlates" with disease activity. Perhaps "is associated with" is a better claim.

Dear Reviewers,

We appreciate your time and efforts to provide constructive feedback on our manuscript. As per the reviewers' suggestions, we have done a major revision of our manuscript. We have marked the newly added and corrected texts in red fonts and underlined them in the original manuscript. By responding to questions and comments from the reviewers, we believe the work presented has been significantly improved. The following is a point-by-point overview of our changes:

Reviewer #1 (Remarks to the Author):

The authors analyzed circulating major lymphocyte subsets in 18 BCR-UV using flow cytometry and found that frequency of NK cells in PBMC was significantly higher in BCR-UV compared to that of healthy controls. Among the NK cells of BCR-UV, CD56dim CD16+NK cells were higher, whereas CD56bright CD16-NK cells were lower than those of controls. Using single cell RNA sequence analysis on 12 BCR-UV and 12 healthy controls, the authors classified NK cells into 10 NK subclusters and found that the cluster 1 was less frequent and clusters 2, 6, 10 were more frequent in BCR-UV. As CD244 and CD8A showed relatively higher expression on the clusters 2, 6, 10, the authors analyzed these molecules on NK cells using flow cytometry and confirmed that CD8 and CD244-expressing NK cells are more frequent in BCR-UV. Furthermore, the expression of CD8 and CD244 were decreased after immunomodulatory therapy. This study is noteworthy in that advanced techniques such as single cell RNA sequence and high-dimension flow cytometry were used to understand immune profiles of peripheral blood of BCR-UV. The finding that CD56dim NK cells are enriched in PBMC of BCR-UV is interesting. CD8+CD244+NK cells were more frequent in active BCR-UV and the NK subset decreased after immunomodulatory treatment are intriguing. However, there are many concerns in the concept of NK cell subsets and the data interpretation which require more improvements.

Major comments

1. Although BCR-UV is strongly associated with HLA-A*29, HLA-A*29 is not recognized by NK cell receptors, which is distinct from other uveitis associated with HLA class I. CD244 and the ligand, CD48, are not involved in HLA recognition. Although CD8 is a co-receptor for TCR, the recognition is not HLA allele-specific. Therefore, the findings in this study are not much associated with mechanisms of "MHC-I -opathies". The authors should make this point clearer in the manuscript.

Our answer: We agree with the reviewer and have removed the statements regarding "MHC-I -opathies" from the abstract and the manuscript. This better reflects the mechanisms described in our work and we hope the reviewer agrees.

2. The authors often used NK1, NK2 as CD56dimCD16, CD56brightCD16-, respectively, however, it is confusing. NK1 and NK2 were previously proposed as Th1 and Th2-like NK

cells in some researchers, and recently, one group named “NK1 and NK2” as “CD56dim and CD56bright NK cells”, respectively, however, these are not widely accepted. The reviewer would suggest not to use NK1/NK2. The authors should refer right references if the authors continue to use the terminology, since the reference referred does not describe anything about NK1/NK2.

Our answer: We agree with the reviewer and have removed NK1/NK2 altogether from the revised manuscript.

3. NK8 is also not well recognized as CD8+ NK cells in this field and the function in NK cells is not well established. In page 14, line 5 from the bottom, “NK8+ cells are considered to be functionally distinct and produce more IFN-gamma and TNF-alpha compared to CD8– NK cells” is an observation in HIV patients. More discussion with previous studies on CD8+ NK cells are necessary.

Our answer: We agree with the reviewer and have changed the term “NK8” to “CD8+ NK cells”. Furthermore, we have included the following paragraphs with more discussion with previous studies on CD8+ NK cells:

pages 9, lines 230-239:

A recent study by McKinney et al. identified CD8+ NK cells in multiple sclerosis (MS) patients that were associated with a more favorable clinical outcome (PMID: 33504809). We compared the hallmark genes from the CD8+ NK cells identified in their study (77 genes) with the expression profile of the CD8+ NK cells identified in this study but did not find significant overlapping genes (2 overlapping genes, Supplementary Fig. 4G). This analysis suggests that the CD244+ CD8+ NK cells found in BCR patients represent a different population as compared to those reported by McKinney et al. (PMID: 33504809). We noted that CD8+ NK cells are found in both the CD56dimCD16+ (approx. 40%) and CD56brightCD16- (approx. 20%) NK subsets (Supplementary Fig. 4H), which may in part explain these differences.

pages 11, lines 285-303

CD8+ NK cells are considered to be functionally distinct and produce more IFN-gamma and TNF-alpha compared to CD8– NK cells (PMID: 25122796). Accordingly, our data show that the CD8+ NK cells-enriched population in patients secretes more IFN-gamma and TNF-alpha compared to healthy controls. In addition, the activation marker CD69 was induced upon treatment with IL-15 or IL-18 (Fig. 3F). This is significant because cross-linking of CD8 on NK cells induces increased expression of the activation marker CD69 at the cell surface (PMID: 16236125). Of interest, elevated frequency of peripheral blood CD8+ NK cells with homing marker CXCR3 is associated with an increased risk for Type 1 diabetes, a T-cell mediated autoimmune condition (PMID: 31130961). The most expanded CD8+ NK subpopulation identified by scRNAseq (cluster 10) in BCR patients was also characterized by high CXCR3 expression (Supplementary Fig. 3D). Whether this CD8+ NK subset (i.e., cluster 10) may directly contribute to eye inflammation in BCR-UV, remains to be determined. Alternatively, the CD8+ NK skewing may be a reflection of diminished regulatory capacity in the NK cell

compartment. NK cells are required to suppress CD8+ T cell autoimmunity (PMID: 32117809) which is attributed to the negative immunoregulation of activated T cells by CD56bright NK cells (PMID: 16585503). As shown in this study, CD56bright NK cells are diminished in circulation of NIU patients, including BCR-UV patients. This makes it tempting to speculate that proinflammatory skewing of the NK cell population diminishes control of autoreactive CD8+ T cell immunity directed towards the eye in BCR-UV.

4. Which data showed that the NK8+-enriched population in patients secretes more IFN-gamma and TNF-alpha compared to healthy controls as described in p14, line 4 from the bottom?

Our answer: We agree that this could be better clarified. We have replotted the data from 5 healthy controls (HC) and 5 uveitis patients (UV) for IFN-gamma and TNF-alpha expression within the CD8+ NK cell compartment below. The data shows significantly high production of IFN-gamma and TNF-alpha by UV CD8+ NK cells upon stimulation. We added these data in more detail to Fig 3G that show that CD8+ NK cells of patients secrete more IFN-gamma and TNF-alpha compared to healthy controls. We hope this answers the question from the reviewer.

We have also sorted CD8+ and CD8- CD16+ NK cells from three healthy donors and *in vitro* stimulated them with golgi-stop for four hours and compared their intracellular cytokine levels by flow cytometry. We don't observe any difference in their cytokines or effector molecules production profiles.

5. The authors showed that CD69 was induced upon treatment with IL-15 or IL-18 on bulk NK cells (Fig. 3F), but not show that for CD8+ NK cells.

Our answer: We agree. We have repeated the experiment with PBMCs from three different healthy donors by treating cells with different dilutions of IL-15, IL-18 and CXCL-16 (control) for 48 hours. We then gated on the CD8+ NK cells and checked the expression of CD69. We have

observed a consistent increase of CD69 level in CD8+ NK cells of all three donors when treated with IL-15 and IL-18, but no expression of CD69 was observed with CXCL-16 treatment. Interestingly, both IL-15 and IL-18 cytokines even in the highest dilutions showed considerable elevation of CD69 level indicating these cytokines are highly potent to induce activation of CD8+ NK cells (data is shown below as histogram).

We have also repeated the experiment with dilutions of IL-15 for additional two donors and looked at CD69 expression in the CD8+ compartment of NK cells (data is shown below as dot plot).

Collectively, this supports our claim that CD69 is induced upon cytokine stimulation on CD8+ NK cells. We have added the new histogram data of IL-15 and CXCL-16 (IL-18 data is not shown, since there was no change of CD69 levels with dilutions) as Supplementary Figure 4D and described it to the results section on page 8, lines 200-206. We hope the reviewer agrees we now conclusively show this for CD8+ NK cells.

6. CD244 is known to hold both activating and inhibitory dual functions depending on conditions, such as expression levels of CD244 and CD48 and higher expression of CD244 could result in inhibitory signal. It is not clear whether CD244+CD8+NK cells in active BCR-UV patient are really activated and involved in the inflammation in BCR-UV. In vitro stimulation of bulk PBMC using PMA/Ionomycin, IL-15, 18 are not enough to show that the subset in the patients is activated. As NKG2D and NKp30 are commonly expressed on NK cells, these expressions do not mean they are activated. The authors should show expression of some activation markers or activating functions of CD244+CD8+NK cells.

Our answer: We agree with the reviewer that additional activation markers would strengthen the claim that CD244+CD8+ NK cells are activated in patients. We also agree with the reviewer that NKG2D and NKp30 are commonly expressed in NK cells and have been shown to increase expression during NK cell activation (PMID: 26203083, PMID: 27933717, PMID: 31921133). We first reanalyzed data for these activation markers NKG2D and NKp30 and observed a significant increase of both markers within the CD244+CD8+ NK cell population of BCR-UV patients, compared to the healthy controls. We added these data to the supplementary data (Fig. S4E, F) and discussed these data in page 8 lines 208-215.

In addition, we also reanalyzed our single cell transcriptomic dataset for transcripts related to inflammation. There are a total of 9619 NK cells (C4_Natural Killer cells) in our dataset. Of the 9619 cells, 296 express at least 1 transcript copy of CD244 and CD8A, which we considered collectively as CD244+CD8+ NK cells. Looking into the transcriptome of these 296 CD244+CD8A+ cells, we found significantly elevated expression of a panel of inflammatory NK genes, including *KLRK1*, *KLRF1*, *CD226*, *PLCG2*, *CSF2*, *TNF*, and *TBX21* in BCR-UV, compared to the healthy controls. We have included the data as supplementary figure S4G. These results have been added to the result section on page 8, lines 216-224.

7. The authors showed changes in frequency of CD8+CD244+NK cells by the treatment course. How about the changes in CD56^{bright}/dim NK cells populations after treatment?

Our answer: Following the question of the reviewer, we reanalyzed the flow cytometry data to look for changes in CD56^{bright}/dim NK cells populations after treatment. We observed a moderate increase in frequency of CD56^{bright} and concomitant decrease in frequency of CD56^{dim} population in some patients during the course of treatment; however, these changes were not statistically significant and majority of the patients did not show any change. These

results were added to supplementary figure S5C and described in the results section page 9-10, lines 253-257.

8. The authors mentioned that flow cytometry was conducted 1-3 days after staining. However, fluorescent signal is getting weaker by time. It is not very reliable if data is acquired after more than one day even though they are fixed. The authors should show the date of flow cytometry analysis for each sample. Were samples from patients and healthy controls analyzed together?

Our answer: As the reviewer points out, previous studies have shown that the time between measurement and antibody staining may influence flow cytometry. We anticipated this in the design of the study by comparing stained samples from the same donor acquired within 24 hours of staining with then after 72 hours of staining. After fixation of cells, our flow-panel shows consistent performance as shown in the example donor below. Furthermore, we plotted the time to acquisition against the frequency of CD8+CD244 and NKG2D+NKp30+ NK cells for each sample, which indicated that time to acquisition did not affect the results. As per the suggestion of the reviewer, we have included the dates to acquisition of each sample (Supplementary Table 1)

9. The authors showed that NK cells of BCR-UV patients produced significantly higher proinflammatory cytokines and suggested that it was due to altered NK1/NK2 balance (reduced CD56bright NK cells) (page 9, line 10). However, it is not correct as CD56bright NK cells can produce higher amount of proinflammatory cytokines. The authors should show the capability of proinflammatory cytokine production with CD56 expression to

demonstrate if ratio of CD56bright/dim affected the differences in responsiveness between BCR-UV and healthy controls.

Our answer: CD56bright NK cells have been reported to be potent cytokine secreting cells, as the reviewer noted. In order to validate this claim, we compared IFN γ and TNF α production by CD56bright and CD56dim populations after stimulation with a leukocyte activation cocktail. In our study, CD56bright and dim NK populations produced comparable levels of IFN γ and TNF α in response to stimulation. Moreover, CD56^{dim} NK cells from uveitis patients did not produce significantly more of these cytokines compared to the controls (as shown in the supplementary figure S1G). The reviewer will find these observations further detailed in the manuscript on page 5, lines 120-126. We hope that the data we presented support our statement.

10. In P9, the 2nd paragraph, the authors concluded that “these results indicate expansion of activated CD8+ NK1 subpopulation characterized by high levels of CD244-signaling molecules in the blood of patients with BCR-UV”. However, frequencies of CD8+, CD244+ NK cells among the clusters 2,6,10 were only around 20% in figure 1D. It is not convincing to say that “the expanded NK1 cluster 0 was defined by high co-expression of CD8 and CD244, in line with our scRNA-seq data” (page9 bottom line).

Our answer: We agree with the reviewer, therefore the statement comparing "in line with our scRNA seq data" was removed from this sentence.

11. The frequencies of each NK cluster classified by flow cytometry should be shown.

We have now added the frequency of the different NK cell clusters identified by flow cytometry in the supplementary table 6 (as shown below).

NK Subclusters	0	1	2	3	4	5	6	7	8	9	10	11
HC 1	3.5	35.4	0.02	7.24	2.26	3.62	10.4	0.36	1.48	1.28	1.92	3.74
HC 2	9.72	31.7	0.92	16.3	3.08	1.24	5.82	0	1.6	1.44	0	0
HC 3	8.6	32.5	0.06	1.84	1.04	0.92	10.1	0.06	5.42	5.6	1.3	1.64
HC 4	2.75	25.6	0.25	8	4.17	4.32	8.88	0.22	5.25	0.18	6.28	2.5
HC 5	3.78	29.8	0	3.94	3.52	0.64	26	0	14.8	0.88	0	0
HC 6	1.6	40.6	0.18	7.14	4.04	0.46	8.56	0.54	14.5	3.84	0.02	0
HC 7	1.14	11.9	0.06	1.66	0.54	2.28	2.4	56.8	1.22	10.7	3.22	7.44
HC 8	11.4	38.1	0.38	2.4	1.18	1.32	9.04	0.04	3.54	0.18	15	17.5
HC 9	2.02	44.6	0	4.72	2.26	0.98	2.1	0.74	1	18.6	3.8	19.2
HC 10	3.5	32.5	0.38	29.1	8.38	16.6	2.62	0	3.38	0.12	0.12	0.12
HC 11	1.58	41.5	0.04	7.78	2.82	3.34	10.3	0.16	1.36	2.26	10.3	16.6
BCR-UV 1	23.4	30.4	0.56	3.08	1.34	0.68	19.4	0.04	7.38	0.1	6.62	6.98
BCR-UV 2	9.06	11.5	0.24	7.32	0.62	0.62	4.58	0.04	0.82	0.06	1.4	2.44
BCR-UV 3	6.64	29.2	0.08	1.4	0.8	0.98	14.8	0	19.7	0.2	6.98	2.96
BCR-UV 4	27.3	16.8	0.22	4.6	1	0.54	35.7	0.04	7.34	0.28	3.04	3.12
BCR-UV 5	30	28.3	1.56	9.74	1.88	1.36	16.3	0	10.5	0.02	0.28	0.08
BCR-UV 6	14	30.1	1.14	3.26	0.7	0.56	10.6	0.1	5.9	0.48	12	20.2
BCR-UV 7	15.6	51.7	2.24	5.96	1.54	1.36	7.86	0	7.76	0.16	1.8	4.1
BCR-UV 8	17.9	22.9	1.68	9	1.56	1.3	14.1	0.08	6.04	0.02	11.8	13.5
BCR-UV 9	19	32.7	1.26	4.3	2.18	1.32	9.4	0.26	5.18	0.1	9.2	15.1
BCR-UV 10	3.48	53.6	0.08	7	2.94	1.06	14.9	0.18	1.84	0.64	2.7	4.64
BCR-UV 11	9.92	42.9	0.12	2.78	0.6	1.78	9.94	0.3	2.68	2.62	5.12	12.5
BCR-UV 12	4.76	35	0	1.34	0.8	0.64	19.3	0.1	6.3	2.84	14.5	11
BCR-UV 13	25.4	38.7	0.34	6.14	2.86	0.46	12.7	0.04	3.32	1.66	0.02	0.04
BCR-UV 14	7.68	25.1	0.24	2.3	0.84	0.52	13.3	0	5.18	2	11.4	8
BCR-UV 15	10.2	21.6	0.16	2.9	0.26	0.16	3.12	0.02	10.1	1.12	1.94	3.52
BCR-UV 16	2.36	28.9	0.12	1.9	0.7	0.44	11	0.02	9.02	0.32	5.9	5.54
BCR-UV 17	4.08	25.8	0.02	1.3	0.56	0.74	35.8	0.06	12.9	16.7	1.58	0.5
BCR-UV 18	0.66	13.3	0.02	0.98	0.26	1.78	0.36	0.18	0.24	75.8	0.7	5.74

12. Where are the data indicating “Differential cluster abundance analysis revealed that clusters 4 and 5 were significantly reduced in the BCR-UV” in page 9, line 4 from the bottom?

Our answer: The reviewer might have overlooked the differential cluster abundance data in Figure 3B. In this graph, the log2 fold change and log10 p-values are shown for all clusters - including clusters 4 and 5. As a clarification, we have also included the following scatter plots of the frequency of cells of clusters 0 to 11 in peripheral blood in BCR-UV versus healthy controls showing significant reduction of clusters 4 and 5 in the patients (Supplementary figure S4C).

13. In Fig 3F, upregulation of CD244 and CD8a by IL15 or IL-18 seems not be concentration dependent. Furthermore, it is generally known that stimulation via single signal is not enough to activate NK cells except for via CD16. How many donors were analyzed?

Our answer: In this figure, a representative graph was shown from three healthy donors. CD244 and CD8a levels were not affected by cytokine concentrations in the experiments, as the reviewer observed. For a more accurate representation of the experiment, we have re-prepared the graph to show the effect of treatment with cytokines in NK cells from 3 donors (Figure 3F).

14. In Fig.3G, it is critical to analyze patients and controls together to compare the CD244 and CD8a expression. Please clarify the date of analysis.

Our answer: We agree with the reviewer that more clarification is required here. To this end, we have prepared the data below, showing 5 representative patients and 5 representative controls that were analyzed together. The data clearly show that BCR-UV patients got more CD8a-bright and CD244-bright NK cells compared to the healthy counterparts.

Because of the confusion created by individual CD8a and CD244 marker analysis and splitting them into high, medium and low expression groups, we reanalyzed the data and compared expression of CD8+CD244+ NK cells in all BCR-UV patients and controls (controls $n=10$, patients $n=18$). We have observed a significant increase in the double-positive NK population in the patient blood compared to the control. The new data is presented now as Fig. 3G.

We have added the dates for the analysis in the supplementary table 1. Our hope is that the reviewer believes the manuscript has adequately addressed this issue.

15. The details about the systemic immunomodulatory therapy conducted on BCR-UV patients should be described.

Our answer: We thank the reviewer for this suggestion, we have included a detailed table (Supplement Table 7) indicating individual patient's treatment category, clinical and anographic activity as well as the systemic immunomodulatory therapy conducted.

16. The authors suggested that clusters 1,6,10 uniquely expressed MYOM2, IGFBP7, LINC00996, respectively. Are these gene expression previously reported in NK cells anywhere?

Our answer: We used the ABIS database to check the expression of these genes across several peripheral blood immune cell populations (Monaco et al. Cell Rep. 2019, available via: <https://giannimonaco.shinyapps.io/ABIS/>). All three genes are expressed in NK cells, as shown in the figure below. This data is included as supplementary figure S3C and described in the result section, page 6, lines 156-160.

<https://giannimonaco.shinyapps.io/ABIS/>

In addition, *MYOM2* and *IGFBP7* (as well as other genes identified in our study, such as *SH2D1B*) are top marker genes for NK cell subpopulations that correlate with disease severity in COVID-19 (PMID: 35501841). Furthermore, *MYOM2* has also been identified in NK cell subset in single cell transcriptomics in tuberculosis, and Alzheimer disease (PMID: 32114394, PMID: 36405736). The discussion section on pages 11, lines 303-310 now includes these details.

17. The abstract contains several issues. CD8A is not the HLA class I restricted antigen. The authors do not show evidence or references that IGFBP7, MYOM2, and LINC00996 are high cytotoxic signatures.

Our answer: We thank the reviewer for pointing this out, we have corrected the abstract.

18. The authors should show evidence or references that SH2D2A is a signaling molecule downstream of CD244.

Our answer: We thank the reviewer for pointing this out. The existing literature suggest that SH2D1A (SAP) and SH2D1B (EAT2) are SH2-domain containing signaling molecules that function downstream of CD244 signaling and promote NK cells activation (reviewed in PMID: 30546369, PMID: 23248626). SH2D2A (TSA_d) is also preferentially expressed in activated T cells and natural killer (NK) cells (PMID: 9468509, PMID: 10587356). This protein has been found to be involved in multiple signaling pathways, including those of the T-cell receptor (PMID: 10587356, PMID: 10553045, PMID: 16446380), however, it's role downstream of CD244 has not been explored yet. We have re-wrote the sentence and corrected the original text (page 11 lines 310-316).

Minor comments

1. In introduction, it is not correct that CD56^{dim} NK cells produce greater amounts of pro-inflammatory cytokines (P3, line 21). It is well known that CD56^{bright} produce more pro-inflammatory cytokines depending on stimulations.

Our answer: Thanks to the reviewer for pointing this out, we have corrected the sentence in the original text.

2. Figure 1A should show a real FSC/SSC facs plot instead of the schema. Furthermore, the leukocytes with highest SSC/FSC should be granulocytes. Macrophages are not present in peripheral blood.

Our answer: We thank the reviewer for suggesting this, we have included a real FSC/SSC FACS plot and corrected the name of the suggested population.

3. What does WDS mean in figure 1C?

Our answer: WDS stands for White Dot Syndromes; also VKH stands for Vogt-Koyanagi-Harada disease. We have included the expansion of these acronyms in the figure legend.

4. Statistical methods should be described in figure 1F.

Our answer: We have described the statistical method for figure 1F in the figure legend.

5. In page 6, line 8 from the bottom, why KLRF1 is not included as the most upregulated genes in BCR-UV?

Our answer: We apologize for missing this important marker and thank the reviewer for pointing this out. We have included this marker in the original text.

6. Please highlight TNFRSF18 (also known as GITR) in the respective figure (P9, line 13). It is hard to find the gene in the figures.

Our answer: We have highlighted TNFRSF18 in supplementary figure S3D, among top 20 gene candidates expressed in subcluster 10.

7. The authors suggested that the cluster 1 classified by scRNA seq had features identical to CD56brightCD16- (p9, line 2), however, the cluster expressed CD16 in Fig 2D.

Our answer: We agree with the reviewer that cluster 1 shows expression of FCGR3A transcripts, as shown in figure 2D. While CD56dim population is often referred to be CD16++, CD56bright population are also referred to as CD16± (PMID: 19278419), apparently because the CD56bright population also express low level of CD16. Therefore, it is not unusual that our protein markers-based defined population CD56brightCD16- NK cells express low levels of FCGR3A transcripts expression.

8. In page 9, line 5 from the bottom, “The remaining 10 clusters in both scRNAseq and flow cytometry analyses are CD56dim and CD16+ populations.” is not correct. The flow cytometry cluster 2 lacks CD16.

Our answer: We thank the reviewer for pointing this out, we have corrected this to ‘predominantly CD16+ with the exception of cluster 2’

9. In p14, line 9 from the bottom, “CD56brightCD16+ NK subsets” should be CD56brightCD16- NK subsets.

Our answer: Thanks to the reviewer for noticing the typo, we have corrected the error.

10. P17, line 1, “healthy controls are collected” should be “were collected”.

Our answer: We have corrected this sentence.

11. Why both RBC lysis buffer (BioLegend #420391) and ACK lysis buffer (Lonza #BP10-548E) were used?

Our answer: We thank the reviewer for pointing out the repetition of RBC lysis protocol. We have used RBC lysis buffer (BioLegend #420391) in this study and corrected the sentence in the revised text.

12. “Pe/Cy7” should be “PE/Cy7”.

Our answer: We have corrected Pe/Cy7 to PE/Cy7

13. “LSRFortessa” should be LSR Fortessa (p18, line 6).

Our answer: We have corrected LSRFortessa to LSR Fortessa

14. Is it correct that the Fc block was conducted in 100% FBS? (P18, line 14) If so, why?

Our answer: The correct one is 100 ul FACS buffer, we have corrected the sentence in the revised text. Thanks to the reviewer for noticing the typo.

15. Reference 19 lacks the page number.

Our answer: We have included the electronic page number (e2016580118) of Reference 19.

16. Fig S1A staining panels should divide into 2 panels, lymphocytes and monocyte/DC panels. CD1c is lacking in the list.

Our answer: We thank the reviewer for this suggestion and have divided the staining panel into two panels- lymphocytes and monocytes/DCs. The markers used in the monocytes/Dc panel are indicated and include CD1c.

17. Statistical methods and P values should be shown in Fig 3H, Fig S1F and S5.

Our answer: We thank the reviewer for pointing out the missing information, we have included the P value and statistical methods in the revised manuscript.

18. Plot for SH2D1B is duplicated in Fig S3B.

Our answer: Thanks to the reviewer for noticing this duplication, we have removed the duplicated plot from the figure.

19. The differences between active and quiet stage are not clear in the fundus photography shown in Fig.S5A.

Our answer: We agree with the reviewer. The extensive birdshot lesions observed in patient's fundus before treatment were resolved over the period of time while the patient was under treatments. To represent the difference between active and quiet stage, we have re-prepared the figure with an expanded portion of the fundus pictures where the lesions before treatment are now clearly visible.

Reviewer #2 (Remarks to the Author):

In this manuscript, the authors performed a transcriptome and FACS analysis on PBMCs from non-infectious uveitis patients and healthy controls (HCs), in order to identify altered NK cell subsets. In the transcriptome analysis, they describe 4 subsets of NK cells (out of 12) with significantly different proportions between birdshot chorioretinopathy patients (a rare form of uveitis patients) and HCs, and one of these may correspond with previously described NK8+ cells, as claimed by the authors. By FACS analysis, the authors describe an accumulation of CD8+CD244+CD16+ NK cell subset in peripheral blood of birdshot chorioretinopathy patients.

The data are of interest, novel and the experiments are well performed. Although the disease is a rare condition, the data are of interest in the field of autoimmune diseases. Fundamentally, the findings add to the complexity of NK cell subpopulations and illustrate the importance of a better phenotyping of NK cells and cytotoxic cells in general.

Furthermore, all illustrations, including the supplementary data, are well presented. The inclusion of age matched controls and the presentation of part of the data with strictly matched controls is appreciated.

Our response: We thank the reviewer for recognizing the importance of our study and also thank for the appreciation of illustrations, supplementary data and inclusion of the controls. We are grateful to the reviewer for critical review that improved the overall quality of the manuscript

A major concern relates to the actual function of the NK8+ cells in uveitis(the authors claim a proinflammatory function in the pathogenesis).

Other comments are minor but need to be addressed.

NK8+ cells

Considering the CD8+ NK cell cluster 0 (as identified by FACS), the authors claim that cluster 10 in the transcriptome analysis corresponds with the NK8+ cells described by others. However, from the heat map shown in figure S3C, there is – to my opinion - no cluster with an increased CD8a expression. Therefore, it will be important that the authors verify whether uveitis patients have an NK8+ subpopulation which correspond with the NK8+ cells described in for example multiple sclerosis patients, described by McKinney et al., Nat Commun 2021, or other available data.

Our answer: We thank the reviewer for pointing this out. In our single cell transcriptional analysis, we have found significant elevation of NK subclusters 2, 6 and 10 in BCR-UV patients compared to healthy controls. All these 3 subclusters have low *NCAM1* (CD56) and high *FCGR3A* (CD16) levels and also express *CD8A* (Fig. 2D and Supplementary Fig. 3C) and we have indicated all these 3 subclusters as CD8+ CD56dim NK cells (which may correspond to cluster 0 in the flow cytometry analysis). However, *CD8A* expression is not limited only to these

subclusters, instead other subclusters except subclusters 3, 9 and 11 also express the transcript.

As per reviewer's suggestion, we have compared top 77 gene expressed in CD8+ NK cells in BCR-UV (NEI-UV), healthy control (NEI-HC) and NK8+ population described by McKinney et al, Nature Com 2021(McKinney-NK8+). While 11 genes overlap between NEI-UV and NEI-HC, we have not found any sharing of the gene signature of McKinney-NK8+ with our CD8+ NK cells. This analysis suggests that the NK8+ cells found in multiple sclerosis patients are different from the CD8+ NK cells found in uveitis patients. We have included the data as supplementary figure 4H and discussed it in page 9, lines 230-239.

Furthermore, the manuscript of McKinney et al. presented evidence for an autoregulatory role of NK8+ cells, by inhibiting activity of potentially pathogenic CD4 T cells, while in the discussion the authors claim that the NK8+ cells found in uveitis patients are pro-inflammatory. It will be important to provide more in vitro evidence for this statement.

Our answer: NK8+ population as described in multiple sclerosis patients by McKinney et al. do not transcriptionally align with our CD8+ NK cells, suggesting that these are different NK populations. While we stimulated the NK cells *in vitro*, we found that CD8+ NK cells from BCR-UV are more responsive in producing cytokines (Fig. 3G). Upon cytokine stimulation *in vitro*, CD8+ NK cells express more CD69 indicating elevated activation state of these cells (Supplementary Fig. 4D). In addition, we have compared the transcriptional profile between BCR-UV and HC and found that CD8+ NK cells from BCR-UV express more activation and inflammation markers than HC (Supplementary Fig. 4G). Taken together, the CD8+ NK cells in BCR-UV patients are likely to be more active and inflammatory than HC.

Minor comments:

Introduction: besides stating that different NK cell clusters have been described in blood of healthy controls, it should also be pointed out that there are tissue-specific NK cells (ILC1 cells) with a phenotype specificity.

Our answer: We thank the reviewer for this suggestion, we have included the following text in the introduction, page 3, lines 63-70:

“NK cells are the major innate lymphoid cells (ILC) that mediate cytotoxicity as well as immune-modulation by producing significant amounts of inflammatory cytokines including interferon-gamma (IFN- γ) and tumor necrosis factor-alpha (TNF- α). NK cells can be broadly divided into two categories; Conventional NK cells (cNK) are located in bone marrow and in peripheral blood. Tissue-resident NK cells (trNK) are found in lymph nodes, the thymus, the lung, the liver, the small intestine, and the uterus, and are considered developmentally and functionally distinct from cNKs.”

Results: scRNA-seq: It is better to write 12 patients and 12 HC, instead of 24 patients and HC.

Our answer: We agree and have corrected this in the text.

Legend Figure 1: Specify the P values show in panel E

Our answer: Thanks to the reviewer for pointing this out, we have included P values in the figure legends.

Legend Figure S1:

Panel A. Specify the lineage (light green).

Our answer: Lineage markers included T cells (CD3), B cells (CD19) and NK cells (CD56) markers and are indicated in the figure legend.

Panel F, phenotyping NK2, include CD16- (also in de figure itself).

Our answer: Thanks to the reviewer for suggesting it, we have corrected it in both figure and in legend.

Legend Figure 2 and S2:

A total of 306,425 cells: please include "PBMC" in stead of cells

Our answer: We thank the reviewer for pointing these out, we have corrected Cells to PBMCs in both the figures and figure legends.

Legend Figure S4:

Panel C & D: Red dotted line captures the region where NK cells are present in BCR-UV, and "relatively" absent in HC.

Our answer: Thanks to the reviewer for suggesting to correct the sentence. Figure S4 C and D suggested an increase of CD244+ and CD8a+ population in BCR-UV, which were relatively absent in HC. However, we realize that the indicated figures were a repetition of what we have

already been showing in Fig. 2B, C, D and E. Therefore, we decided to remove the repetitive figures and replace them with more informative ones.

Figure 3:

Beside plotting cluster 0, as significantly different between HC and Uveitis patients (panel C), it would be good to also plot clusters 4 and 5 (or include FACS data of all clusters in a suppl figure).

Our answer: In response to the reviewer's suggestion, we now added the frequency of each cluster indicated in Fig. 3A as Supplementary figure 4C and compared the differences between HC and BCR-UV.

Panel F: regarding the in vitro stimulation of NK cells, was this not with purified NK cells (or total PBMCs and gated on NK), and with cells obtained from HCs? This should better be explained in the legend and corresponding results section.

Our answer: In Fig. 3F, NK cells were purified from three different healthy donors and stimulated with indicated cytokines for 48 hours. We have added this information in the figure legend and in the corresponding result section.

Figure 4:

Panel B: it is mentioned that MFIs histograms are plotted from patients at base, Month 6 and Year1, together with HCs, but HCs are missing in this panel.

Our answer: We thank the reviewer for pointing this out. We have included the MFI of CD8a and CD244 from a corresponding healthy control in the revised figure 4B. The frequency analysis of CD8+CD244+ NK cells in Fig. 4C and D also includes data from corresponding healthy controls.

Reviewer #3 (Remarks to the Author):

This is an interesting manuscript which presents intriguing data on a possible role for a rare subset of NK cells in the pathology of uveitis; most specifically, the rare birdshot chorioretinopathy.

Our response: We thank the reviewer for finding our work interesting and for providing us with the critical reviews. We have carefully gone through each and every concern raised, below. We hope that the reviewer agrees that the points have been addressed well.

I have a major issue with the use of the term NK1 vs NK2 throughout the paper. These subsets are not determined by the levels of expression of CD56 and CD16 as implied throughout and the reference #16 is not correct to justify this use of terminology (line 72). Cytokine secretion profile and CD95 expression are the relevant criteria for NK1 vs NK2. CD56++/CD16- are cytokine secreting, poorly cytotoxic NK cells which are considered to be immature. For clarity, I would edit line 78 to:

"This includes CD56dim CD16+ NK subsets which may be "inflammatory", identified by secretion of high levels of cytokine and interferon response genes.

NK1 and NK2 cells were shown by Deniz et al (2002) to have similar cytolytic function but to be separated on the basis of the degree of Ifn- γ secretion. Both subsets expressed CD16.

Our answer: We agree with the reviewer's comment and removed the terms NK1 and NK2 entirely from the manuscript and rather defined the subsets as CD56dimCD16+ and CD56bright populations. We have also edited line 78 as suggested by the reviewer.

The techniques used are highly appropriate for the study and are well described. However, the choice of statistical tests is not discussed nor justified and it is difficult to believe that some of the findings reach the degrees of statistical significance which are claimed. For example, Fig 1C presents "percent live" NK cells but it is not stated whether these data are means values with SDs or SEMs or medians with ranges etc etc.

Our answer: We thank the reviewer for this comment and apologize for missing clarification on the statistical methods. In Fig. 1C we used Box and whiskers plot and presented values are medians with ranges. We have done a non-parametric Mann Whitney U test and compared each disease condition with HC. P values presented are **** $P < 0.0001$, *** $P = 0.0002$, ** $P = 0.004$, * $P = 0.01$. We have included the statistical information in each of the figure legends of 1C, 1E and 1F.

The red bar showing the uveitis patient data overlaps more than 50% of the HD population and is highly skewed by the three outlier data points. This does not appear to be a Normal distribution yet an unpaired Student t test has been used to assess statistical significance which is wrong.

Our answer: We thank the reviewer for pointing this out. To confirm the significance test in the current data we have used a non-parametric test Mann Whitney U test:

- Healthy-vs-Uveitis_All: ***, P value 0.0006
- Healthy-vs-WDS: non significant, P value 0.2000
- Healthy-vs-Birdshot: ****, P value <0.0001
- Healthy-vs-VKH: non significant, P value 0.5903
- Healthy-vs-Definite Sarcoidosis: *, P value 0.0352
- Healthy-vs-Presumed Sarcoidosis: non significant, P value 0.8502
- Healthy-vs-Serpiginous: *. P value 0.0236
- Healthy-vs-Undifferentiated: non significant, P value 0.0783

We have corrected the data in the manuscript using the P values from the Mann-Whitney U test.

There are similar issues with the data presented in Fig 3H. Whilst the populations of CD244 expressing NK cells are certainly different between the BCR-UV subjects in the "lo" versus the "mid" and "hi" groups; it is important to know how these were assessed for statistical significance since both the "mid" and "hi" groups are bivariate distributions and neither is a single normal distribution and cannot be tested as such with a parametric test. It looks as though the "mid" and "hi" subjects include two distinct clusters, the lower of which are not significantly different than the "lo" group whereas the upper cluster represents a different type of patient.

Our answer: We thank the reviewer for pointing this out. We have redone the statistical analysis using non-parametric Kruskal-Wallis test with Dunn's multiple comparison and could not find any significant difference between HC and BCR-UV in 'mid' or 'hi' population of the corresponding markers.

We then compared CD8+CD244+ NK cell population between HC and BCR-UV and found a significant increase of the double-positive cells in BCR-UV than the HC. We therefore, replaced the old figure with the new data.

The Methods section needs a detailed explanation and justification of the statistical tests used.

Our answer: We thank the reviewer for this suggestion, we have included the following in the Methods section on page 16-17, lines 460-469:

“Statistical analysis

Quantification and data analysis of experiments are expressed either as median with ranges (box and whiskers plot) or as mean \pm standard deviation (bar and dot plots). *P* values were calculated using analysis of variance (ANOVA- Tukey’s multiple comparison test) or two-tailed Student’s t-test for pairwise comparisons or non-parametric Mann-Whitney U test or Kruskal-Wallis test with Dunn’s multiple comparison and were calculated using Graphpad Prism v.9. Differences in proportions of scRNAseq data between the groups were assessed using the “scProportionTest” package⁵⁵, with number of permutations set at 10,000. Qualitative experiments were repeated independently to confirm accuracy.”

On a more prosaic point, it is unclear what parent population has been used for the "% live" determination. Are these data showing the percentage of live NK cells within the live lymphocytes or total PBMC?

Our answer: We thank the reviewer for this comment, for clarification we have included the following para in the Method under Flow cytometry section on page 14, lines 393-398:

“Flow cytometry gating was done as- Lymphocytes (SSC-A vs FSC-A)/Single Cells (FSC-H vs FSC-A)/Single Cells (SSC-H vs SSC-A)/Live (Aqua L/D vs FAS-A). CD20- (CD20 vs FSC-A)/CD56+ (CD56 vs CD3) cells are gated as NK cells (Supplementary Fig. 1E). Data shown % Live NK cells are the frequency of CD56+ cells within Live gated cells.”

It is also important to know whether these differences between the percentages of NK subsets in patients versus HD are reflected in the absolute numbers of NK cells per ml of peripheral blood.

Our answer: We share the reviewer’s concern. Unfortunately, we have not included the count beads in samples during the acquisition. However, we have always processed the same

volume of fresh blood (3 mL; please see in Method: *Blood Sample Processing* and *Flow Cytometry*) for flow staining, reconstituted 3 million of PBMCs into 300 uL FACS buffer and flow-acquired 200 uL of cells in from all patients and healthy control samples (please see Method: *Flow Cytometry*). Therefore, the difference in frequency we observed in the immune cell population would represent the absolute numbers and can be compared among individuals.

The data showing changes in CD244+/CD8+ NK cell percentages during treatment are very interesting but too limited to justify the claim in line 336 that this cell population "correlates" with disease activity. Perhaps "is associated with" is a better claim.

Our answer: We thank the reviewer for suggesting the more appropriate way to present our observation with the disease activity and CD244+/CD8+ NK cell percentages. We have corrected the sentence as suggested (page 12, line 327).

REVIEWER COMMENTS

Reviewer #1 (Remarks to the Author):

The manuscript has been improved. However, there are still several concerns remain.

1. On page 5, paragraph 2,
Capabilities of cytokine production by CD56bright and CD56dim depend on the type of stimulation. Only comparing the differences by PMA/Ionomycin stimulation (the leukocyte activation cocktail the author used) cannot draw a clear conclusion on the association of CD56 expression with NK cell response in BCR-UV.
2. PMA/Ionomycin often reduces cell surface markers, which may fail to distinguish CD56dim and CD56bright NK cells. The authors should show actual facs plots before and after stimulation with PMA/Ionomycin with intracellular staining conducted in Fig1F.
3. The comparison between BCR-UV and healthy controls in Fig 1F needs to be more convincing as the authors used PBMC for BCR-UV and NK cells for healthy controls, which may affect the results.

Minor points

1. Why was the Kruskal-Wallis test used in Fig3H, which compared only 2 groups?
2. Why was the unpaired t-test used in Suppl Fig.5B, which compared the change of CD8/CD244 expression levels in BCR-UV by time?
3. Through the manuscript, +/- such as in CD16+, CD16-, CD8+ should be superscript.
4. In the discussion (page 10, line 279), the role of CD8 in HLA recognition by NK cells still needs to be better understood. Please discuss this point.
5. The activating and inhibitory dual nature of CD244 should also be discussed in the manuscript.

Reviewer #2 (Remarks to the Author):

The authors have addressed all of my comments and I especially appreciate the reanalysis of their data regarding CD8+ NK cells described in MS and the CD244+ NK cells.

I believe that the CD8+ NK cell population will be a topic of major interest in other

autoimmune and auto-inflammatory diseases.

Reviewer #3 (Remarks to the Author):

Thank you for your considered and comprehensive responses to my initial comments. I am happy that my concerns regarding the tests of statistical significance have been addressed suitably.

REVIEWER COMMENTS

Reviewer #1 (Remarks to the Author):

The manuscript has been improved. However, there are still several concerns remain.

1. On page 5, paragraph 2, Capabilities of cytokine production by CD56bright and CD56dim depend on the type of stimulation. Only comparing the differences by PMA/Ionomycin stimulation (the leukocyte activation cocktail the author used) cannot draw a clear conclusion on the association of CD56 expression with NK cell response in BCR-UV.

Our answer: The reviewer correctly pointed out that the Leukocyte Activation Cocktail is a polyclonal cell activation mixture containing the phorbol ester PMA and a calcium ionophore. Polyclonal activators like PMA/Ionomycin, concanavalin A, lipopolysaccharide, phytohaemagglutinin, staphylococcus enterotoxin B, and monoclonal antibodies directed against subunits of the TCR/CD3 complex are particularly useful for inducing cytokine-producing cells. Since our target cells are NK cells, we used the Leukocyte Activation Cocktail to induce cytokine response. We agree with the reviewer that a different type of stimulation, like a receptor-ligand interaction, may result in differential expression of cytokines in NK cell subsets. For clarifying, we have included the following sentence (page 5, lines 129-132):

“The polyclonal stimulation did not affect the production of cytokines by CD56bright and CD56dim NK cells. However, these subsets may differ depending on the type of stimulation used.”

2. PMA/Ionomycin often reduces cell surface markers, which may fail to distinguish CD56dim and CD56bright NK cells. The authors should show actual facts plots before and after stimulation with PMA/Ionomycin with intracellular staining conducted in Fig1F.

Our answer: We agree with the reviewer that PMA/Ionomycin stimulation could influence expression of certain cell surface markers. However, we have not found any differences in expression levels for the markers CD45, CD3 or CD56 staining after stimulation with the Leukocyte Activation Cocktail; we could gate CD45+CD3-CD56+ NK cells and also could distinguish between CD56dim and CD56bright NK cells after stimulation. As per the reviewer's request we have prepared below the FACS plots with gating of CD45, CD56 and CD3 before and after stimulation for all eight samples used in Fig. 1F.

3. The comparison between BCR-UV and healthy controls in Fig 1F needs to be more convincing as the authors used PBMC for BCR-UV and NK cells for healthy controls, which may affect the results.

Our answer: We thank the reviewer for pointing this out and apologize for the typographical error. PBMCs from both BCR-UV and HC were stimulated with the

Lymphocyte Activation Cocktail and analysis of the cytokine expression was done after flow-gating on NK cells. We have corrected the sentence to “Analysis was conducted by stimulating PBMCs and flow-gating on NK cells from BCR-UV (red) and healthy controls (HC, blue).”

Minor points

1. Why was the Kruskal-Wallis test used in Fig3H, which compared only 2 groups?

Our answer: We agree with the reviewer that Fig. 3H is comparing between two groups of normally distributed samples, therefore Kruskal-Wallis test is inappropriate. We have done Student's t test and updated the figure and its legend.

2. Why was the unpaired t-test used in Suppl Fig.5B, which compared the change of CD8/CD244 expression levels in BCR-UV by time?

Our answer: In Suppl Fig. 5B, we don't have adequate sample numbers for Month 03 and Month 06 groups and we have not included them in the statistical comparison. We have compared between two groups, Baseline vs Year 01, and therefore used the t test for statistical analysis. We have updated the figure legend to clarify this point.

3. Through the manuscript, +/- such as in CD16+, CD16-, CD8+ should be superscript.

Our answer: We thank the reviewer for this suggestion, we have corrected +/- to superscript throughout the manuscript.

4. In the discussion (page 10, line 279), the role of CD8 in HLA recognition by NK cells still needs to be better understood. Please discuss this point.

Our answer: We thank the reviewer for this suggestion. We hope the following addition to the discussion will help to better understand the role of CD8 in HLA recognition by NK cells (page 11, lines 288-297):

“In T cells, CD8 receptor helps recognition of the specific MHC-peptide (pMHC) complex by the T cell antigen receptor (TCR). pMHC-I molecules are major ligands for both forms of CD8 and the binding affinity of CD8 α /pMHC is comparable to that of CD8 $\alpha\beta$ (PMID: 8955273, PMID: 18025211). Similar to CD8 $\alpha\beta$ a co-receptor for TCR, CD8 α also function as a co-receptor for KIRs in NK cells and enhances pHLA-I binding to, for example KIR3DL1, in an expression-dependent manner (PMID: 31420518). Moreover, CD8 α also binds with MHC class I molecules expressed on NK cells in the NK–NK synapse that leads to the rescue of CD8+ NK cells from secretion-induced apoptosis- a mechanism that involves CD8-induced influx of calcium from the extracellular sources (PMID: 16236125).”

5. The activating and inhibitory dual nature of CD244 should also be discussed in the manuscript.

Our answer: We have added additional discussion on the dual nature of CD244 (Page 12, lines 335-351):

“CD244 was first discovered in NK cells and CD8+ T cells as a stimulatory cell surface receptor that mediated non-MHC-restricted killing by NK cells and CD8+ T cells (PMID: 8228228, PMID: 17111350, PMID: 11714782). However, later it was suggested that CD244-deficient murine NK cells were more cytotoxic than wild-type NK cells (PMID: 15607804). A model proposed based on the relevant discoveries suggests that CD244 can transmit intracellular signal either through activation or inhibitory intracellular signaling proteins (PMID: 17100873). For instance, the binding of CD244 to SLAM-associated protein (SAP) upregulates the viability and cytotoxic effects of NK cells and CD8+ T cells, but its binding to Ewing's sarcoma-activated transcript-2 (EAT2) or EAT-2-related transducer (ERT) may transmit inhibitory signals (PMID: 17100873). Since mature human NK cells abundantly express SAP (PMID: 11917118), the role of CD244 in human NK cells appears to have activating functions. However, mature mouse NK cells express SAP, EAT-2, and ERT (PMID: 16127454) leading the receptor to contribute to either stimulation or inhibition NK cell activation (PMID: 17100873).”

REVIEWERS' COMMENTS

Reviewer #1 (Remarks to the Author):

The manuscript was improved according to the suggestions.